# *k*-Pareto Optimality-Based Sorting with Maximization of Choice and Its Application to Genetic Optimization

Jean Ruppert [1,*,†], Marharyta Aleksandrova [2,*,†] and Thomas Engel [2]

1   Mathematics and Computing S.à r.l., L-2360 Luxembourg, Luxembourg
2   Department of Computer Science, Faculty of Science, Technology and Medicine, University of Luxembourg; 2 Avenue de l'Universite, L-4365 Esch-sur-Alzette, Luxembourg
*   Correspondence: jean.ruppert@mathcomp.lu (J.R.); marharyta.aleksandrova@gmail.com or marharyta.aleksandrova@uni.lu (M.A.)
†   These authors contributed equally to this work.

**Abstract:** Deterioration of the searchability of Pareto dominance-based, many-objective evolutionary optimization algorithms is a well-known problem. Alternative solutions, such as scalarization-based and indicator-based approaches, have been proposed in the literature. However, Pareto dominance-based algorithms are still widely used. In this paper, we propose to redefine the calculation of Pareto-dominance. Instead of assigning solutions to non-dominated fronts, they are ranked according to the measure of dominating solutions referred to as *k*-Pareto optimality. In the case of probability measures, such re-definition results in an elegant and fast approximate procedure. Through experimental results on the many-objective 0/1 knapsack problem, we demonstrate the advantages of the proposed approach: (1) the approximate calculation procedure is much faster than the standard sorting by Pareto dominance; (2) it allows for achieving higher hypervolume values for both multi-objective (two objectives) and many-objective (25 objectives) optimization; (3) in the case of many-objective optimization, the increased ability to differentiate between solutions results in a better compared to NSGA-II and NSGA-III. Apart from the numerical improvements, the probabilistic procedure can be considered as a linear extension of multidimentional topological sorting. It produces almost no ties and, as opposed to other popular linear extensions, has an intuitive interpretation.

**Keywords:** genetic algorithms; multi-objective optimization; topological sorting; linear extension of multidimensional sorting

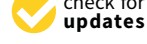



## 1. Introduction

Genetic algorithms are a group of biologically-inspired search optimization techniques. Due to the flexible definition of basic elements, such as the population of individuals, fitness, cross-over, mutation, and selection, these algorithms can solve optimization problems of different natures. Genetic optimization was successfully used for task scheduling in cloud computing [1], for performance improvements in recommendation algorithms [2], and in the production industry [3], to name a few. If the number of objectives is larger than one, a solution of such an optimization problem consists of a set of non-dominated points, called a Pareto set, which defines a Pareto-frontier in the space of objectives. In this case, the goal of the optimization algorithm is to generate a set of solutions situated as close as possible to the true Pareto-frontier, maintaining a high diversity at the same time.

Optimization problems with two or three objectives are known as *multi-objective* problems; in the case of higher dimensionality, they are referred to as *many-objective* problems [4]. Several algorithms exist to effectively solve multi-objective optimization problems, among them PESA-II [5], NSGA-II [6], and SPEA2 [7]. It was shown that the non-dominated sorting procedure based on the Pareto dominance relationship is very effective in this

case [8]. However, it is also known that Pareto dominance-based *many*-objective optimization evolutionary algorithms face various difficulties. Among them, the deterioration in the searchability of Pareto dominance-based sorting [4,9]. As stated in [10], this happens due to the lack of selection pressure. Indeed, when the number of objectives increases, the number of incomparable or equally preferable solutions grows exponentially.

To overcome this problem, a number of alternative approaches were proposed. Among them, there are relaxed dominance-based approaches [11], the reduction in the number of objectives via scalarization [12], and the introduction of additional indicators to guide the selection process [13], such as hypervolume. These approaches provide some advantages; however, they also have disadvantages. In particular, the indicator-based approaches require calculating the value of the the relative indicator function; scalarization-based methods require either running several single-objective optimizations during one run or running individual single-objective optimizations over many runs [12]; finally, as stated in [9,11], the diversity maintenance can become more difficult in relaxed dominance-based approaches.

In this paper, we study in detail the application of the recently introduced *k*-Pareto optimality-based sorting [14] to the problem of *many*-objective genetic optimization. Here, *k* stands for the value of the Pareto optimality. In the rest of the text, we use the terms *k*-Pareto optimality and Pareto optimality, as well as the corresponding abbreviations *k-PO* and *PO*, interchangeably.

It was shown theoretically that this sorting procedure maximizes the choice, or diversity, of a subset of the best elements selected with respect to a dominance relation. This characteristic can be valuable for genetic optimization, as a diverse gene pool allows exploring the search space more efficiently. In [14], we suggested to sort solutions by *k*-Pareto optimality instead of the traditional Pareto dominance and presented preliminary results demonstrating the high potential of this approach. This paper is an extension of the above-mentioned study with respect to genetic optimization. It presents more detailed experimental results through the following contributions:

1.  Based on the proposed ranking metric *Pareto-optimality*, we study three genetic algorithms in detail: *PO-count*, *PO-prob*, and *PO-prob\**; see Section 4 for the detailed description. All three algorithms are based on *NSGA-II* and differ from the latter in the selection procedure by:

    -   *PO-count*: counting *PO*, which consists of counting the number of dominating solutions (counting *PO*);
    -   *PO-prob*: approximating *PO* via a probabilistic procedure (probabilistic *PO*);
    -   *PO-prob\**: sequentially combining probabilistic calculation of *PO* and Pareto dominance sorting from *NSGA-II*.

2.  We compare the proposed methods with *NSGA-II* and *NSGA-III* using the 0/1 knapsack problem with the number of objectives varying from 2 to 25. Our experimental results with random and tournament selection show the following:

    -   Ranking by counting *PO* provides almost identical results as ranking by Pareto dominance;
    -   Using probabilistic *PO* ranking allows us to increase the hypervolume of the resulting solutions for many-objective and most multi-objective optimization problems;
    -   With the increase in the number of the objectives, probabilistic *PO* yields a set of solutions that are almost never dominated by the solutions of other algorithms;
    -   In general, *PO-prob* and *NSGA-III* algorithms yield fewer extreme solutions when the number of objectives increases;
    -   We demonstrate that probabilistic *PO* ranking is computationally much more efficient. It allows for reducing the time complexity of the sorting procedure from $O(N^2M)$ to $O(NMlog(N))$, where $N$ is the population size and $M$ is the number of objectives.

3.  Our algorithms are implemented as an extension of the Python library for evolutionary computation DEAP (https://deap.readthedocs.io/en/master/, accessed on

28 January 2022). They are available as open source (https://github.com/marharyta-aleksandrova/kPO, accessed on 28 January 2022).

4.  We also discuss how Pareto optimality can contribute to the problem of interpretability, see Section 6.

The rest of the paper is organized as follows. In Section 2, we discuss the related state-of-the-art research work. Section 3 presents the proposed *PO*-based sorting and the related algorithms. In Section 4, we describe experimental setup, and in Section 5, we analyze the characteristics and the performance of the proposed algorithms via experiments. Finally, Sections 6 and 7 contain concluding remarks and a discussion of possible future work.

## 2. Related Work

In this section, we review the relevant state-of-the-art research works. We discuss the approaches that aim to solve the deterioration of the searchability problem, see Section 2.1, and the computational complexity of genetic algorithms, see Section 2.2.

### 2.1. Searchability Deterioration of Pareto Dominance-Based Sorting

As it was stated in the previous section, the Pareto dominance-based selection criterion does not provide enough selection pressure to guide the evolution process in the case of many-objective optimization [4,9,10]. Numerous attempts have been made to overcome this problem. We classify and schematically represent them in Figure 1.

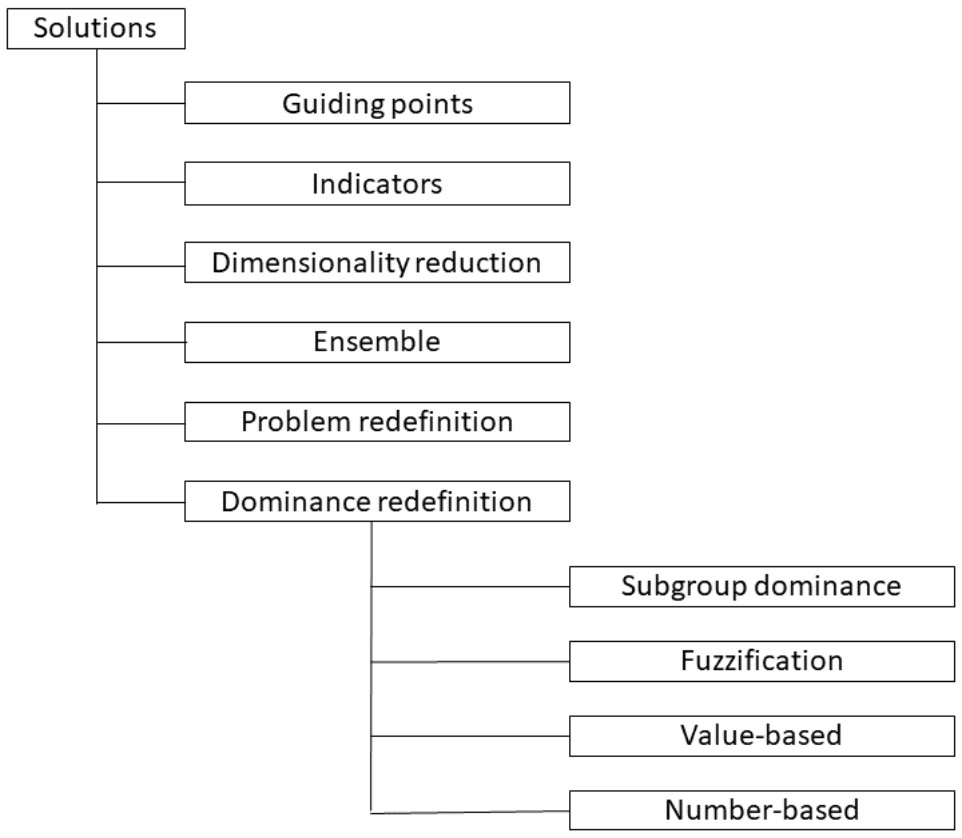

**Figure 1.** Solutions for the searchability deterioration problem.

To the first group of approaches, we classify those that require specific **points** to guide the selection process. These can be knee-points [9,15] and reference points either provided by a user or identified automatically, for example, the reference points in *NSGA-III* [16]. Apart from directing the selection process towards the true Pareto-frontier, these reference points also contribute to the diversity of the solution set as they are often widely distributed in the search space.

The next group of methods employs **indicators** to select the best solutions. Among recent works in this area, we can mention [10,17], which incorporate several ranking methods simultaneously, and [18] which used a novel convergence indicator. Computing the values of these indicators increases the execution time of the optimization process.

Another group of approaches aims to solve the root of the problem by **reducing the dimensionality**. This can be achieved, for example, by using a transfer matrix [19] or by considering single-objective optimization problems via scalarization [12]. Such methods simplify complex relationships between objectives. This can lead to under-coverage of the search space.

There was also a suggestion to **ensemble** several methods. In the approach from [20], one population is evolved using conventional non-dominated sorting, and another is evolved using an approximate non-dominated sorting procedure. This combination is supposed to improve both diversity and convergence at the same time.

An alternative approach proposed in [21] performs sorting not with the aim to select the best individuals but to remove the most unfitted ones (**problem redefinition)**. It was implemented within the the framework of *NSGA-III* and is based on the idea of the niche-preservation operation of this algorithm.

The methods from the last group attempt to **re-define** the way the **dominance** is calculated. According to [11], we can define two main subgroups here: *value*-based dominance and *number*-based dominance. Value-based dominance methods modify the Pareto dominance by changing the objective values of the solutions when comparing them, for example, $\epsilon$-dominance [22]. Number-based methods try to compare a solution to another one by counting the number of objectives where it is better than, the same as, or worse than another; $(1-k)$-dominance [23] and *L*-optimality [24] are prominent examples. Alternative methods also include the fuzzification of Pareto-dominance, see [25,26], and the definition of dominance within a subgroup of a population, for example, $\theta$-dominance, which was used to improve the convergence of *NSGA-III* in [27].

The method studied in this work, $k$-Pareto optimality, falls within the subgroup of number-based approaches. As other methods from this group, it is straightforward and easy to implement. It does not require additional constructions such as reference points or indicators. In addition, to the best of our knowledge, it is the only procedure with theoretical guarantees of diversity, see [14]. As the current paper is a proof of concept, we perform empirical comparison of the newly proposed method with the well-established approaches, *NSGA-II* and *NSGA-III*. In future work, we will aim to compare its performance with that of other value- and number-based approaches.

### 2.2. Computational Complexity

In the case of high-dimensional problems, non-dominated sorting not only suffers from the deterioration in the searchability, but it is also slow. A number of attempts were made to improve its time complexity. Some methods are based on **algorithmic improvements**. For example, efficient non-dominated sorting (ENS) [8] reduces the time complexity by comparing solutions only with those that have already been assigned to a front. Similarly, authors of [28] proposed assigning solutions to the fronts in the ascending order of the sum of objectives. There are also approaches that are based on usage of **alternative data structures**, such as trees [29]. The last group of methods performs **approximate** non-dominated sorting. The first algorithm from this group, ANS, was proposed in 2016 [30]. In this algorithm, the dominance relationship between two solutions is determined by a maximum of three objective comparisons on top of a population sorted according to one of the objectives. It was shown by the authors of [30] and further confirmed in other research works [31] that this approach is not only more computationally efficient, but it also leads to better search performance. This idea was further developed in [32], where no more than two objectives were compared.

In this paper, we demonstrate that $k$-Pareto optimality based on probabilistic calculation, the *PO-prob* algorithm, allows for substantially improving the computational efficiency

of genetic optimization, see the results in Section 5.3. The usage of probability theory makes this approach somewhat similar to approximate methods. However, *PO-prob* utilizes the values of all objectives to compare solutions.

## 3. *k*-Pareto Optimality-Based Sorting for Genetic Optimization

In this section, we describe the general framework of genetic optimization, see Section 3.1, and introduce sorting by *k*-Pareto Optimality, see Section 3.2. We also discuss the characteristics of this novel sorting approach that can be useful for genetic optimization and illustrate its differences from the traditional non-dominated sorting, see Section 3.3.

### 3.1. Genetic Optimization Overview

Let us consider $n$ objectives $f_i$, $i \in \{1, \ldots, n\}$, and a maximization task which consists of the simultaneous maximization of all $n$ objectives over a set $X$ of solutions, that is,

$$\text{maximize } \{\vec{f}(a) \equiv (f_1(a), f_2(a), \ldots, f_n(a))\}. \\ \text{over } a \, \in X \tag{1}$$

The binary concept of Pareto-dominance is defined as follows.

**Definition 1.** *An individual a* Pareto-dominates *an individual b if:*
$\forall i \in \{1, \ldots, n\} : f_i(a) \geq f_i(b) \quad \wedge \quad \exists i \in \{1, \ldots, n\} : f_i(a) > f_i(b).$

An individual is said to be *Pareto-optimal* if it is not Pareto-dominated by any other element. The goal of multi-objective optimization is to find a set of Pareto-optimal solutions spread over the whole Pareto-frontier. Genetic optimization algorithms aim to solve this problem based on a biologically inspired heuristic: the evolution of a set of randomly initialized solutions.

In Figure 2, we present the general flowchart of the genetic optimization procedure. The main steps of the procedure are explained below.

1.  First, we create an initial population of solutions (or individuals) of a predefined size, *pop_size*, and evaluate the fitness of every individual with respect to a predefined fitness function, step 1. In the case of multi- or many-objective optimization, this is a function with multi-dimensional output.
2.  Second, we select a subset of individuals for "procreation" based on a chosen parent selection criterion, step 2. Two popular criteria are random and tournament selection. In random selection, the parents are chosen randomly; in tournament selection, the choice is made based on fitness among a set of randomly selected individuals of a predefined size.
3.  Next, on the step 3, the chosen parents are "mated" to create a required number of children or offspring. This operation is also known as crossover. Often, the number of children is equal to the size of the original population. The generated offspring can also be mutated. The latter process is controlled by the value of the mutation probability, *mut_prob*. After evaluating the fitness values for the newly created offspring, the two sets of solutions are joined.
4.  Finally, on the selection step, step 4, the combined population is sorted according to a specific criterion, and the best *pop_size* individuals are advanced to the next generation. The evolution continues until a certain criterion is fulfilled. Often, the process is controlled by the maximum number of generations, *n_gen*.

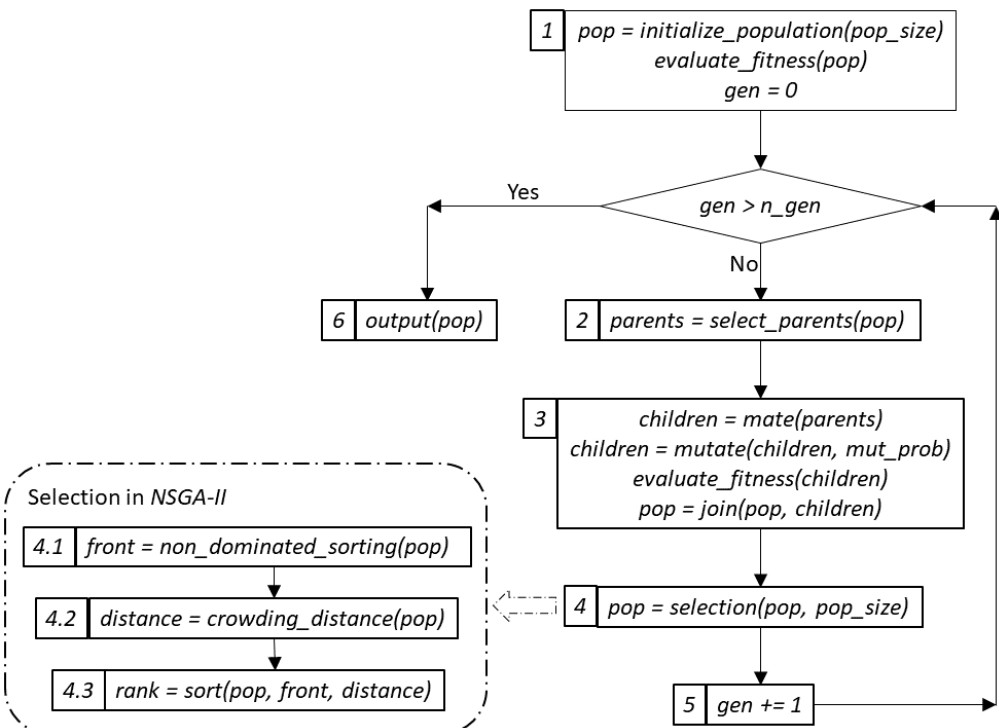

**Figure 2.** Flowchart of the genetic optimization procedure.

When the number of objectives is larger than 1, various methods can be used to sort the combined population of solutions in step 4 of the flowchart. The popular *NSGA-II* algorithm performs the following: First, all solutions are assigned to fronts using the Pareto dominance (*PD*) relation. Next, the crowding distance is calculated for all individuals within the same front. As reflected in the name, the crowding distance measures the distance to the nearest individual within a front. Finally, the population is sorted with respect to fronts and the crowding distance as the primary and the secondary criteria. This procedure ensures that the most fitted solutions advance to the next generation. The usage of crowding distance allows one to choose the solutions that are spread all along the non-dominated front and helps avoid concentration. See [6] for a more detailed presentation of *NSGA-II*.

### 3.2. Formal Definition of k-Pareto Optimality

As we can see, the selection step of a genetic optimization procedure requires sorting the population of solutions according to a predefined dominance relation. In [14], we introduced sorting by *k*-Pareto optimality and demonstrated its utility in many-objective genetic optimization.

Let us consider a set $X$ with a binary relation $R$. In the context of $n$ objectives introduced in the previous section, $R$ corresponds to the relation $aRb$ iff $\forall i \in \{1, \ldots, n\}$ : $f_i(a) \geq f_i(b)$. Thus, Pareto dominance corresponds to the relation $R^*$ defined by $aR^*b$ iff $aRb \wedge \neg(bRa)$. We also consider a positive and $\sigma$-finite measure $\mu$ defined on $X$. The measure $\mu$ can be defined in different ways. Important examples are the counting measure and probability measures. Depending on the definition of $\mu$, it can indicate the following characteristics of the elements in $X$: *how many?*, *how likely?*, *how important?*, etc. Thus, we have a measure space $(X, \Sigma, \mu)$, where $\Sigma$ is a set of subsets of $X$ and $\mu$ intuitively indicates the size of these subsets.

The *k*-Pareto optimality of an individual $a$ is the measure of the set of points that Pareto dominate $a$ and is written as $\mathrm{po}(a)$. Formally, $\mathrm{po}(a) = \mu(\{x : xR^*a\})$. If $\mu$ is a probability measure, the *k*-Pareto optimality of an element $a$, $\mathrm{po}(a)$, is the *likelihood* an element drawn

at random from $X$ is strictly preferable to $a$. In the case of the counting measure, $\mathrm{po}(a)$ is the *number* of elements from $X$ that are strictly preferable to $x$.

Choice between $a$ and $b$ means that neither $a$ Pareto dominates $b$ nor $b$ Pareto dominates $a$. That is, $a$ and $b$ are equally preferable and belong to the same Pareto front. The notion of choice is closely related to the notion of *diversity*. This concept is more practically interesting and can be formally defined as the likelihood that two elements drawn at random from $A$ offer choice. Thus, it is the choice offered by $A$ divided by $\mu(A)^2$.

In [14], we formally proved that among all topological sorting methods, sorting by $k$-Pareto optimality maximizes the diversity of the subset of the best individuals of a pre-defined *measure* (*size* in the case of counting measure). This property holds for any type of element (points on a plane, graphs, functions, etc.), dominance or preference relations (smaller/larger than, parent/child, etc.), and measures defined on the set under investigation (counting measure, probability measure, etc.). The property of diversity maximization suggests that this sorting procedure can be useful in the selection step of genetic optimization. Indeed, the maximization of diversity of the generated solutions allows for investigating the search space more efficiently. Thereby, we suggest changing the non-dominated sorting to the sorting by $k$-Pareto optimality in the step 4.1 of the flowchart presented in Section 2. We also experiment with two types of measures $\mu$: a probability measure and the counting measure.

### 3.3. Illustrative Example

We illustrate the proposed approach and its difference from the traditional non-dominated sorting with the help of an example shown in Figure 3, see [14] for more details and derivations. In this example, the search space consists of 6 points defined on a 2D plane, and the goal is to maximize both objective functions defined by the relative coordinates of the points.

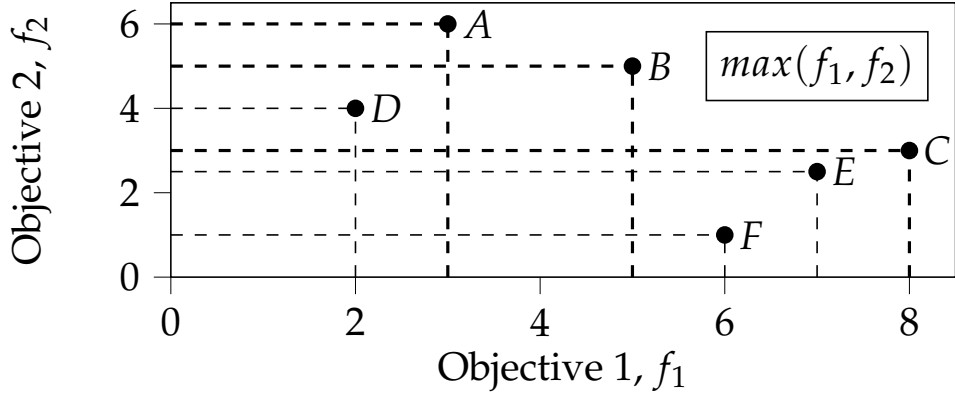

**Figure 3.** Illustration of sorting for a maximization problem.

Let us first consider the traditional non-dominated sorting by Pareto dominance. This sorting procedure distributes the available solutions between Pareto-fronts in the following way: First, all non-dominated points are identified. In the example shown in Figure 3, there are three non-dominated points: $A$, $B$ and $C$. These three points make up the first front. Next, the points of the newly identified front are removed from consideration, and the process is repeated. Continuing this process, we obtain the second front made up of two points, $D$ and $E$, and the third front containing only one point, $F$. The relative distribution of points between Pareto fronts is summarized in the second column of Table 1.

**Table 1.** Distribution of points between fronts for different sorting procedures: *PD*—Pareto dominance; *PO-count*—counting Pareto optimality; *PO-prob*—probabilistic Pareto optimality; *PO-prob*, $\epsilon$—probabilistic Pareto optimality with an adjustment for 0 probability.

| Front | PD | PO-count | | PO-prob | | PO-prob, $\epsilon = 0.1$ | |
|---|---|---|---|---|---|---|---|
| | | Points | Val. | Points | Val. | Points | Val. |
| 1 | A, B, C | A, B, C | 0 | A, C | 0.00 | C | 0.05 |
| 2 | D, E | E | 1 | B | 0.08 | A | 0.07 |
| 3 | F | D, F | 2 | E | 0.11 | B | 0.08 |
| 4 | | | | D, F | 0.28 | E | 0.11 |
| 5 | | | | | | D, F | 0.28 |

Instead of sorting by Pareto dominance, we propose to sort the points by the measure of the dominating solutions, which we refer to as Pareto optimality (*PO*). Let us consider the counting measure. The sorting is thus conducted by the *number* of dominating solutions. In the text, we refer to this procedure of calculating *PO* values as counting *PO*, or *PO-count*, as it implies counting the number of dominating solutions. Similarly to the previous case, no other point dominates the points *A*, *B*, or *C* in Figure 3. Thereby, the value of the counting Pareto optimality for these points is 0. Point *E* is dominated by one point *C*, resulting in the corresponding *PO* value of 1. Finally, points *D* and *F* are dominated by two other points: *A*, *B* and *C*, *E* respectively, meaning that their Pareto optimality is 2. The front of the point is determined by its *PO-count* value, see the third and the fourth columns of Table 1.

Comparing sorting by *PD* and *PO-count*, we can observe that the number of identified fronts is the same: three fronts in total. However, the point *D* moved from the second to the third front when transitioning from Pareto dominance to the counting Pareto optimality-based sorting. Also note that the first front is always the same for both procedures. Indeed, a non-dominated point will always have the value of *PO-count* equal to 0 and vice versa.

Instead of the counting measure, we can also consider a probability measure. We refer to the relevant procedure as probabilistic *PO*, or *PO-prob*. In this case, instead of calculating the number of points that dominate the current one, we can estimate the probability of this point to be dominated. In our calculations, we assume that there are no duplicates. This is also enforced in our experiments. If we have no a priori knowledge on how the objectives are correlated, we assume them to be independent. Thus, the probability of any point to be dominated can be estimated by multiplying the probabilities of being dominated with respect to the individual objectives.

Let us calculate the values of *PO-prob* for the example presented in Section 3 assuming the independence of objectives $f_1$ and $f_2$. Point *A* is never dominated with respect to $f_2$; thereby, the probability of being dominated with respect to this objective is 0, $P\_dom_{f_2}(A) = 0$. Considering the first objective $f_1$, 4 points out of total number of 6 points dominate the point *A*, $P\_dom_{f_1}(A) = 4/6 \approx 0.667$. By multiplying these two values, we can estimate the value of *PO-prob* as $P\_dom_{f_1}(A)P\_dom_{f_2}(A) = 0.667 * 0 = 0$. Let us now analyze in a similar way the point *B*. It is dominated by three points with respect to $f_1$ (points *C*, *E* and *F*), resulting in $P\_dom_{f_1}(B) = 3/6 = 0.5$. Considering the second objective, only point *A* dominates *B*. This gives us $P\_dom_{f_2}(B) = 1/6 \approx 0.167$. Thus, the value of *PO-prob* for point *B* is $P\_dom_{f_1}(B)P\_dom_{f_2}(B) \approx 0.5 * 0.167 \approx 0.083$. The corresponding values for the rest of the points and the distribution of the points between fronts are presented in the sixth and the fifth columns of Table 1 respectively. As we can see, now the 6 points are distributed among 4 fronts, with *B* singled out into a separate front.

A possible drawback of calculating *PO-prob* as explained above can be multiplication by 0 when a point is non-dominated with respect to at least one objective. In this case, the value of *PO-prob* is always equal to 0 no matter what the domination probability with respect to other objectives is. An easy solution to this problem is to replace the value 0 with

a small number $\epsilon$ in the calculations. The last two columns of Table 1 show the respective values of *PO-prob* for $\epsilon = 0.1$. As we can see, now point $C$ is more preferable to $A$, as its domination probability with respect to $f_2$ is less then the domination probability of $A$ with respect to $f_1$. Indeed, $P\_dom_{f_2}(C) = 3/6 = 0.5$ and $P\_dom_{f_1}(A) = 4/6 \approx 0.667$. At the same time, $P\_dom_{f_2}(C) < P\_dom_{f_1}(A)$ and $P\_dom_{f_1}(C) = P\_dom_{f_2}(A) = 0$. In the rest of the text and in our implementation, we use this version of calculating *PO-prob* with $\epsilon = \frac{1}{pop\_size}$, where $pop\_size$ is the population size or the number of analyzed solutions. We refer to it simply as *PO-prob*. If we assume that there are no identical solutions, then *PO-prob* is an estimation of the probability a randomly chosen point *non*-strictly dominates the analyzed one, see [14].

In the case of counting measure, sorting by *PO* is equivalent to sorting by dominance rank relationship [33]. In addition, it was shown in previous works that sorting a random sample of independent continuously distributed points by *PD* and *PO-count* has the same limiting behaviour when the number of points becomes infinitely large [34]. Although *PO-count* was already studied in the literature, *PO-prob* was not. In addition, *PO-prob* is not the only possible algorithm based on Pareto optimality, as similar approaches can be developed for different measures and relations expressing preference. That is why we present evaluation results for both *PO-prob* and *PO-count* for comparison and more detailed analysis.

In the following sections, we evaluate how *PO*-based ranking affects the convergence properties of genetic optimization algorithms.

## 4. Experimental Setup

Our experimental setup is summarized in Table 2. To perform the experimental evaluations, we chose the 0/1 multi-objective knapsack problem with independent objectives as defined in [35]. The mathematical formulation of this problem is presented in Appendix A. As stated in [35], this optimization problem is easy to formulate, but at the same time, it is rather general. It is also representative of a certain class of real-world problems. This problem was used in numerous research papers to compare evolutionary algorithms and to study different aspects of their performance, see [4,36], for example. Finally, it scales easily to higher dimensions, preserving its properties.

**Table 2.** Experimental setup.

| | |
|---|---|
| Problem | 0/1 knapsack, $n_k \in \{2 - 8, 10, 15, 25\}$, 250 items |
| Baseline | *NSGA-II, NSGA-III* |
| Algorithms | *PO-count, PO-prob, PO-prob** |
| Implementation | python package *DEAP* |
| Parent selection | random, binary tournament |
| Crossover | uniform |
| Parameters | $n\_gen = 500$, $pop\_size = 250$, $mut\_prob = 0.01$ |
| Runs | 30 |

We use 10 different test problems with the number of objectives $n\_k \in \{2 - 8, 10, 15, 25\}$ and 250 items. We adopt two selection schemes: random selection and binary tournament selection with replacement. For all algorithms and all settings, we execute genetic optimization for 500 generations. We also use uniform crossover with mutation probability 0.01 and population size 250.

We implement our approach within the framework of the *NSGA-II* algorithm by redefining the sorting procedure using *PO-count* and *PO-prob* as an alternative to *PD*, see step 4.1 in Figure 2. We refer to these algorithms by the name of the chosen sorting method. Additionally, we experiment with a combination of *PO-prob* and the classical *NSGA-II*. In this approach, we run *PO-prob* during the first 350 generations and then switch to *NSGA-II* during the last 150 generations. We refer to the latter approach as *PO-prob**.

As it was shown in Section 3, sorting based on Pareto optimality results in less incomparable solutions (ties). In addition, theoretical results from [14] guarantee maximization of diversity when sorting by *PO*. Thereby, we hypothesize that *PO*-based ranking can produce better results for many-objective problems. To demonstrate this, we compare our approaches with the basic *NSGA-II* algorithm and its modification designed for many-objective optimization, *NSGA-III*. The *NSGA-III* algorithm additionally takes reference points as an input parameter. We use deap.tools.uniform_reference_points function to generate uniformly distributed reference points. As suggested by the authors of *NSGA-III*, the number of reference points is chosen to be close the value of population size, see [16]. We limit the number of baseline approaches to these two classical algorithms because, as stated in [36,37], many new algorithms are overspecialized and can perform poorly in general settings.

We implement the proposed algorithms as an extension to the *DEAP* python library. Unless stated otherwise, the values reported in this paper are averages of 30 independent runs.

## 5. Experimental Results

In this section, we present the results of the experimental evaluation. We discuss the characteristics of the analyzed algorithms in Section 5.1. Next, we proceed to the performance analysis in Section 5.2. Finally, Section 5.3 is dedicated to the time complexity analysis.

### *5.1. Characteristics of the Proposed Approach*

In this subsection, we describe the characteristics of the proposed approaches. In particular, we analyze the distribution of objective function values and the fraction of solutions belonging to the first front. To better understand the behavior of the studied algorithms, we start with a visualization of the evolution.

### 5.1.1. Evolution of Solutions for $n_k = 2$

For a random initialization, we plot the evolution of solutions and the first front of the last generation for $n_k = 2$ in Figure 4 (random selection) and Figure 5 (tournament selection). We choose to use the same initialization for all algorithms to show the difference in the resulting solutions. *NSGA-II* and *NSGA-III* are well-studied, so we concentrate more on the behavior of the algorithms based on *PO*.

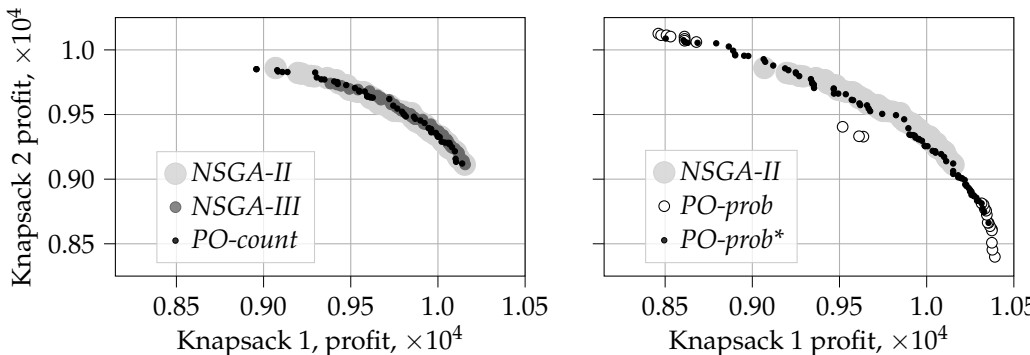

(**a**) First front for *gen* = 500, counting algorithms    (**b**) First front for *gen* = 500, probabilistic algorithms

**Figure 4.** *Cont.*

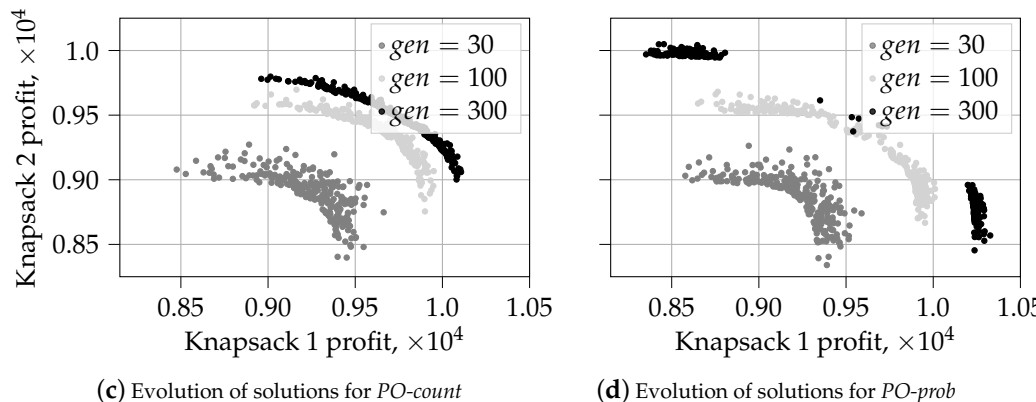

(**c**) Evolution of solutions for *PO-count*

(**d**) Evolution of solutions for *PO-prob*

**Figure 4.** Visualization of solutions for $n_k = 2$ and random selection. *y*-axis is shared among plots.

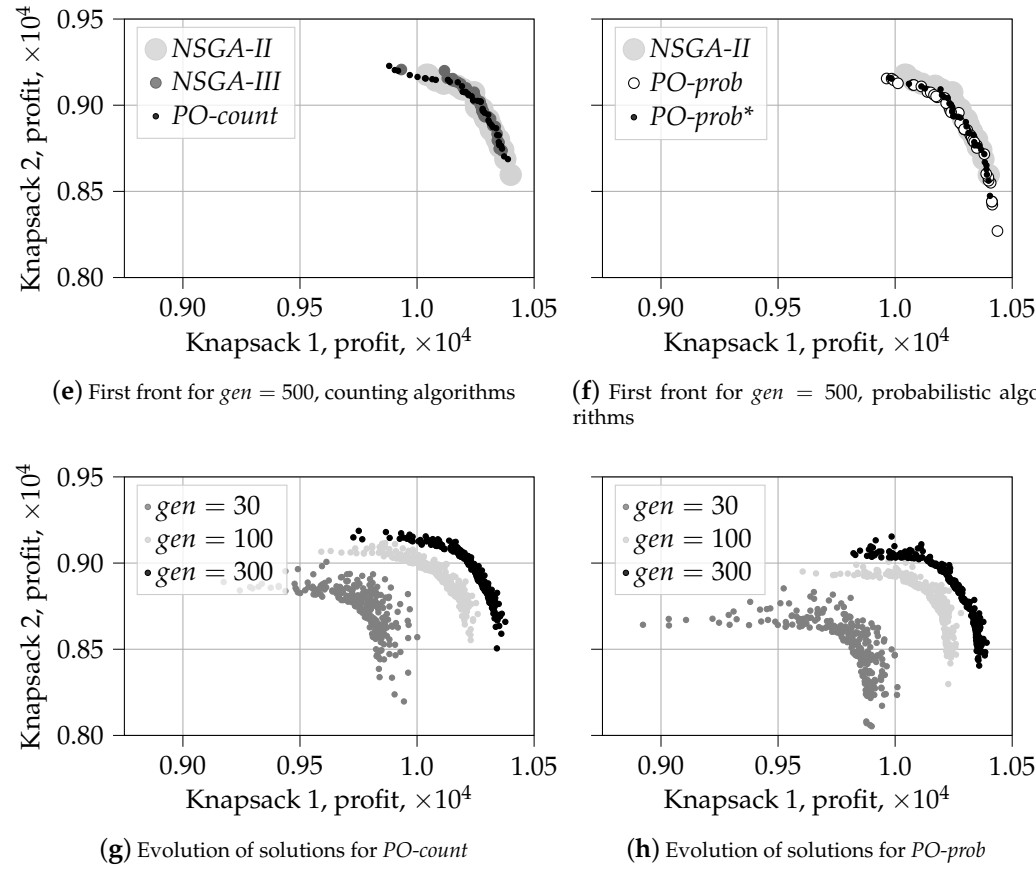

(**e**) First front for *gen* = 500, counting algorithms

(**f**) First front for *gen* = 500, probabilistic algorithms

(**g**) Evolution of solutions for *PO-count*

(**h**) Evolution of solutions for *PO-prob*

**Figure 5.** Visualization of solutions for $n_k = 2$ and tournament selection. *y*-axis is shared among plots.

We start with the analysis of the results for **random selection**. Figures 4c,d show three generations of solutions for *PO-count* and *PO-prob*: *gen* = 30, *gen* = 150, and *gen* = 300. As *PO-prob\** performs identically to *PO-prob* until *gen* reaches 350, its intermediate solutions are not plotted. Figures 4a,b show the first front of the final generation *gen* = 500 for all algorithms.

We can notice interesting behavior of the *PO-prob* algorithm: after a certain number of generations, the produced solutions tend to move to extreme values with very few solutions concentrated in the middle part of the Pareto-frontier, see Figure 4d for *gen* = 300. This pattern was observed in all 30 independent runs. It can be explained by the nature of the probabilistic sorting used in *PO-prob* and by the fact that probabilities are multiplied. Indeed, a solution will be ranked high if its probability to be dominated by any other solution is low. In this case, the extreme solutions have little probability to be dominated

with respect to one of the objectives. This contributes to their higher rankings in comparison to the solutions from the middle of the Pareto-frontier. Thereby, such solutions are advanced to the next generations and have higher probability to pass their characteristics to offspring. At the same time, in the case of *NSGA-II*, all solutions from the first non-dominated front with similar values of crowding distance are equally preferable. This observation is consistent with the computational example presented in Section 3. As we can see from Table 1, the extreme solutions *A* and *B* are more preferable than the solution from the center, *C*, for *PO − prob*-based sorting.

This tendency is, however, "repaired" by *PO-prob\**. The solutions of the latter first reach the extreme values using sorting by *PO-prob* and then cover the whole length of the Pareto-frontier using *PD* sorting from *NSGA-II*. We can also see that continuing evolution with *PO-prob* allows it to reach even more extreme solutions during the last 150 generations.

The solutions in the middle of the Pareto frontier produced by *PO-prob* are clearly dominated by the solutions of other algorithms. At the same time, only *PO-prob\** can reach extreme solutions almost as well as *PO-prob*. Overall, *NSGA-II* covers the middle part of the Pareto frontier better than all other algorithms. *NSGA-III* and *PO-count* demonstrate behavior similar to *NSGA-II*.

In the case of **tournament selection**, *PO-prob* behaves similarly to other algorithms and produces non-fragmented fronts, see Figure 5. At the same time, tournament selection results in worse coverage of the Pareto-frontier than random selection, compare Figures 4 and 5. The maximum achieved values of objective 2 with tournament selection are around $0.92 \times 10^4$. From Figure 4d, we can see that the corresponding values for random selection were achieved for $gen = 100$. At this point, the front is not yet fragmented. This might indicate that extreme and middle solutions are advanced further during different periods in the evolution process, and an adaptive selection procedure might be required for *PO-prob*. We aim to investigate this question further in our future work.

As in the previous case, *PO-prob* covers the extreme solutions better, see Figure 5b. However, a large portion of its solutions is dominated by the solutions produced by other algorithms. The difference between *PO-prob* and *PO-prob\** in this case is less visible. At the same time, using *NSGA-II* during the last 150 generations in *PO-prob\** does push the resulting Pareto-frontier a bit further. From Figure 5a, we can again see that *NSGA-III* and *PO-count* behave very similar to *NSGA-II*. Overall, tournament selection usually favors one objective more than another. These observations were confirmed in a large fraction of our 30 independent runs.

5.1.2. Visualization of the First Front for $n_k > 2$

To analyze the solutions for larger numbers of knapsacks, when $n_k > 2$, we opt for a different type of visualization. Among all 30 independent runs of each algorithm, we choose a single run with the median hypervolume value, following the methodology from [36]. In Figures 6 and 7, we show the solutions from the first front of the last generation for 7 and 25 knapsacks, respectively. The similar visualization for $n_k = 2$ is shown in Appendix B. The latter supports the results presented in Figures 4 and 5 and serves for comparison purposes. The solutions are shown using parallel coordinates with the horizontal axis representing the index of the objective and the vertical axis showing the corresponding objective value. Starting from $n_k = 6$, the distributions of solutions for random selection and tournament selection have similar characteristics. This is why we show results only for random selection.

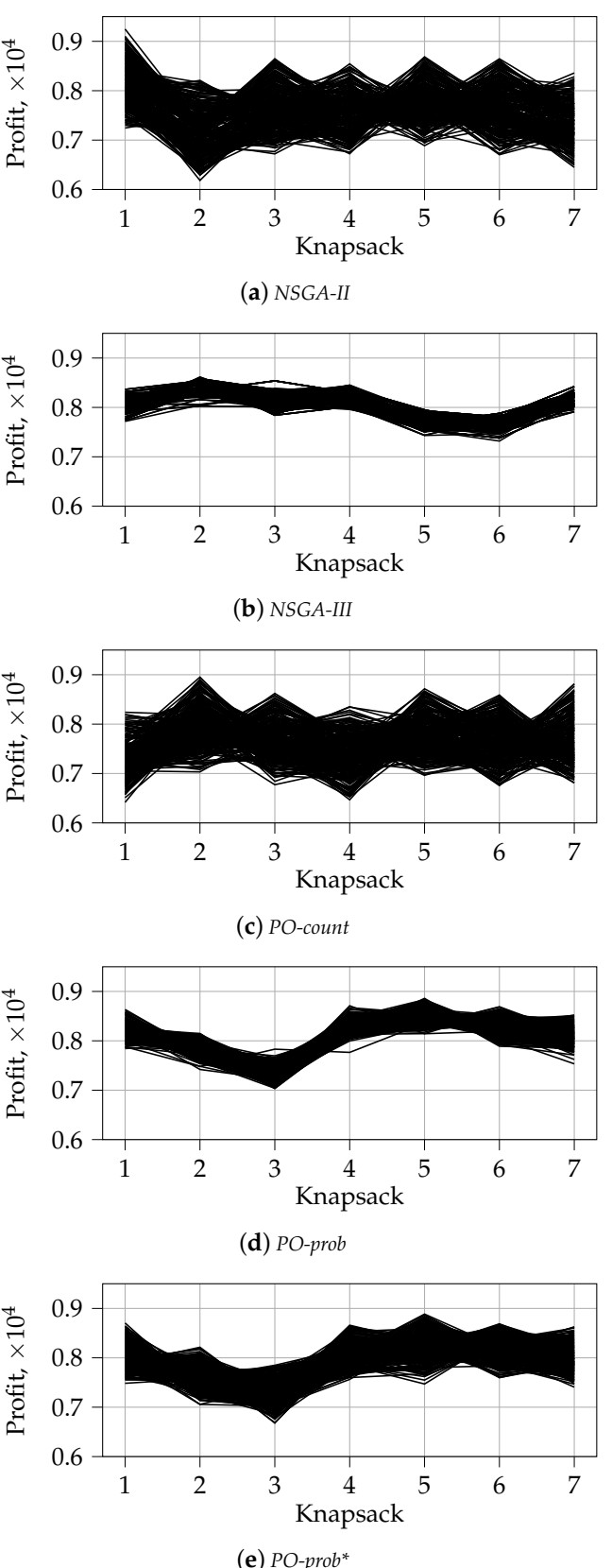

**Figure 6.** Final Pareto frontier for $n_k = 7$ and random selection.

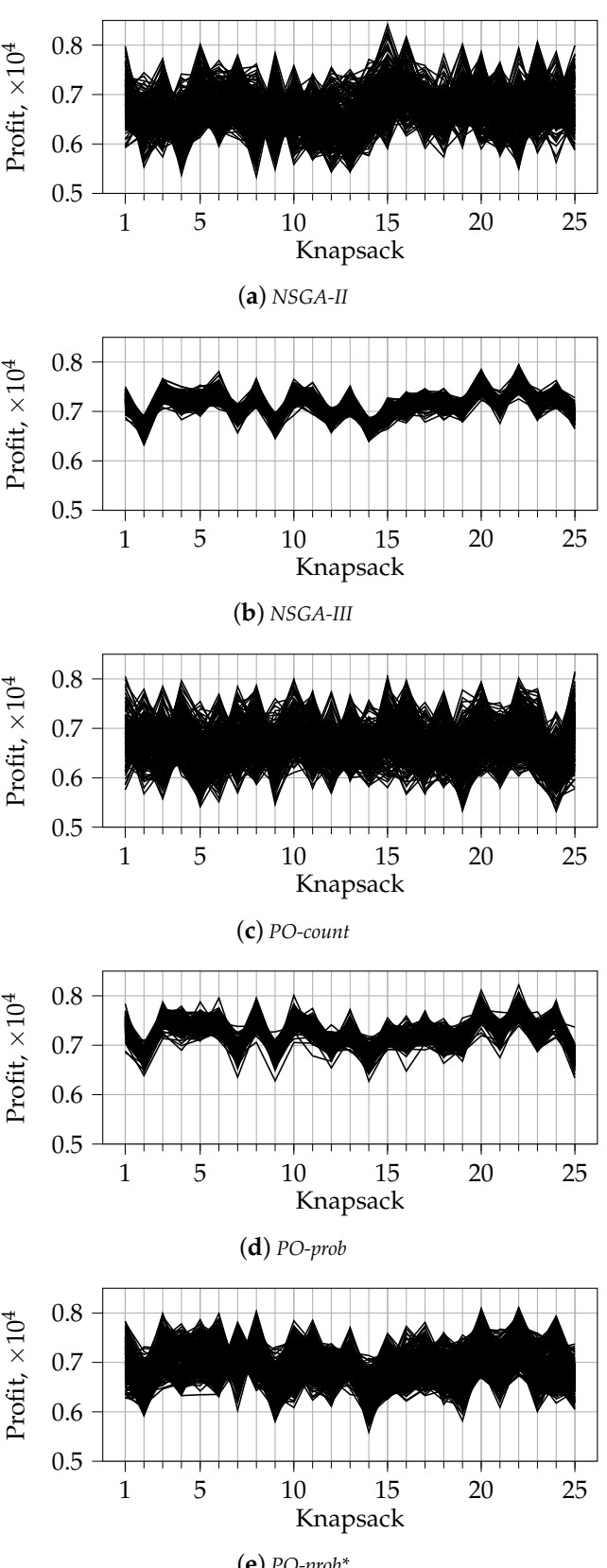

**Figure 7.** Final Pareto frontier for $n_k = 25$ and random selection.

We can notice some differences as compared to the results for $n_k = 2$. In particular, the tendency of tournament selection to prefer one objective more than another disappears for $n_k \geq 6$. This is reflected in the absence of disproportionate values for one of the ob-

jectives in Figures 6 and 7 (compare with the corresponding results for $n_k = 2$ shown in Figures A1 and A2 from Appendix B). Furthermore, we can notice the absence of gaps between the solutions for *PO-prob*. This means that the related fronts are not fragmented, as it was in the case of two knapsacks and random selection, see Figures 4b,d. After examining the results for different numbers of knapsacks, we noticed that this pattern holds true for any number of knapsacks larger than 2, $n_k > 2$.

Finally, we can see that the range of solutions for *NSGA-III* and *PO-prob* is tighter than that for other algorithms. This pattern becomes visible for $n_k = 5$ and holds true for larger values. In addition, the solutions for these algorithms tend to be situated in the middle and the top part of the objective values range: from 6000 to 9000 for $n_k = 7$ and from 5000 to 8000 for $n_k = 25$. This means that, when $n_k$ becomes larger, *PO-prob* produces less extreme solutions than *NSGA-II*. This demonstrates the advantage of the former algorithm in the case of many-objective optimization. *NSGA-II* and *PO-count* produce more extreme solutions than other algorithms. The characteristics of *PO-prob\** are in between those for *PO-prob* and *NSGA-II*. This results naturally from the fact that *PO-prob\** is a combination of these two algorithms.

The results presented in Appendix C compliment the above study of the relative position of solutions in the space by analyzing the average distance to the diagonal of the corresponding hypercube. It supports the findings presented in Section 5.1.1 and the current one.

5.1.3. Fraction of Non-Dominated Solutions

The fraction of the population that belongs to the first front shows how difficult it is for an algorithm to differentiate between the solutions and select the most fitted ones. If all solutions belong to the first front, then they are all incomparable for the given algorithm, in the sense that none of them is preferred over another. The inability of an algorithm to differentiate between solutions indicates that it cannot direct the evolution process.

For the small example in Section 3, we show that sorting by *PO-prob* with $\epsilon > 0$ results in a first front with only one element, and in the lowest number of ties (incomparable solutions) in general. This observation is also supported by our experimental results. Apart from some rare cases, *PO-prob* with $\epsilon = 1/pop\_size$ assigns every solution to a separate front. However, when the population size increases, the probability of having more than one solution per front quickly tends to zero. This is also supported by the results presented in Table 1, *PO-prob* with $\epsilon = 0.1$ also assigns one point per front, with the last front being an exception. *PO-prob* with $\epsilon = 0$ has the same properties with one exception: the number of solutions in the first front is equal to the number of objectives $n_k$. Indeed, in this case, all solutions that are the best according to one of the objectives will have the value of Pareto optimality equal to 0, see the discussion for *PO-prob* in Section 3. This observation holds true for all setups and for every generation. It also demonstrates the ability of *PO-prob* to differentiate between solutions in both multi- and many-objective optimization.

As opposed to *PO-prob*, sorting the given population by *PO-count* and by *PD* always results in the same first front. Indeed, all nondominated solutions have Pareto optimality values equal to 0 and are assigned to the first front, see Section 3. The remaining fronts, however, are typically different. This results in different solutions generated over the evolution process. To illustrate this difference, we show in Figure 8 the fraction of solutions belonging to the first non-dominated front for all studied algorithms. Note, that the first non-dominated front here is identified with sorting by Pareto dominance and is referred to as *first ND-front*. However, every algorithm uses its own sorting procedure to evolve the population. We present results for the following number of knapsacks: $n_k = 2$, $n_k = 6$, and $n_k = 10$. The relative distributions for random and tournament selection have different trends only in the case of two knapsacks. That is why, for $n_k = 6$ and $n_k = 10$, we report the results only for random selection.

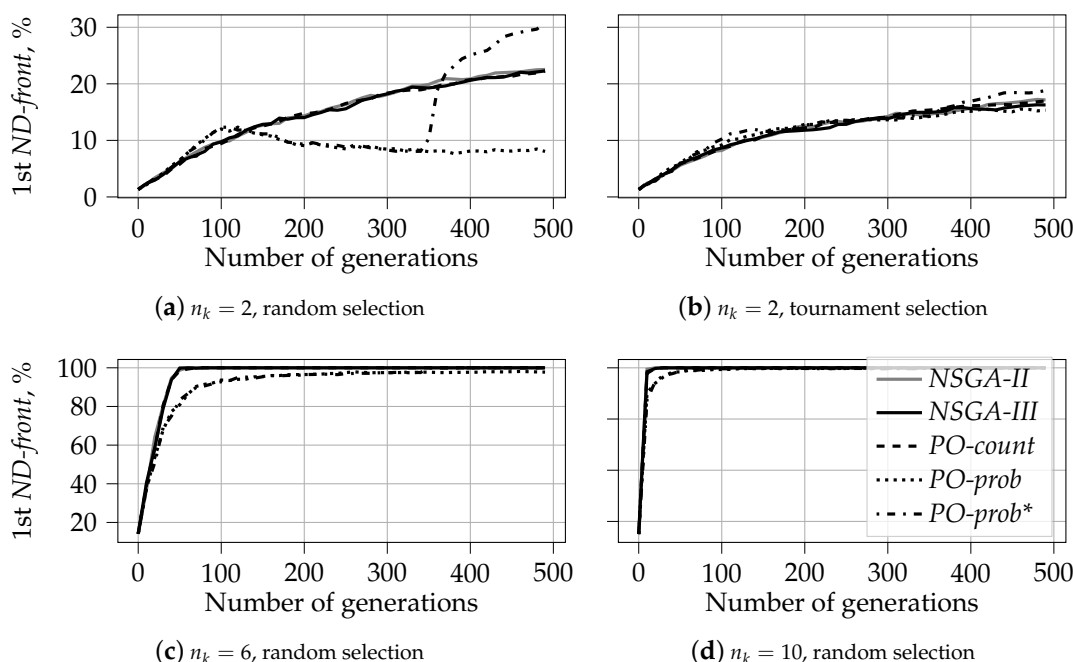

**Figure 8.** Percentage of solutions in the first non-dominated front (sorting by Pareto dominance, *PD*) as a function of number of generations. The legend and *y*-axis are shared among plots.

Figure 8a,b show that for two knapsacks, the size of the first non-dominated front is relatively low for all algorithms. In the case of random selection, see Figure 8a, the maximum value of 30% is reached by *PO-prob\** for *gen_num* close to 500. The fraction of solutions in the fist ND-front is approximately 10% for *PO-prob* and does not exceed 25% for *NSGA-II*, *NSGA-III*, and *PO-count*. The large size of the first non-dominated front produced by *PO-prob\** is due to the combination of extreme solutions constructed by *PO-prob* during the first 350 iterations and solutions spread out in the middle, which are constructed by *NSGA-II* during the last 150 generations, see Figure 4b.

For tournament selection, all algorithms produce the first non-dominated front of almost the same size, see Figure 8b. The maximum value in this case does not exceed 20%. It is visible, however, that the size of the first ND-front is slightly less for *PO-prob* and is slightly larger for *PO-prob\**. This agrees with the results presented in Figure 5b. As before, *PO-prob* identifies more extreme solutions than *NSGA-II*, but this difference is not as large as in the case of random selection, see Figure 4b for comparison.

Analyzing the results for six knapsacks in Figure 8c, we can see that, very quickly, all solutions generated by *NSGA-II*, *NSGA-III*, and *PO-count* start belonging to the first non-dominated front. After approximately 50 generations, all solutions belong to the single first ND-front. At the same time, *PO-prob* produces more than one non-dominated front up to the last generation $gen = 500$. However, the percentage of solutions belonging to the first ND-front is still high: it reaches 90% for $gen = 100$ and reaches its maximum of 97% for $gen = 500$. As expected, *PO-prob\** follows *PO-prob* until $gen = 350$ and joins *NSGA-II* after that.

Figure 8d shows the corresponding results for $n_k = 10$. The fraction of solutions belonging to the first non-dominated front becomes close to 100% very quickly for all algorithms. This happens at $gen = 20$ for *NSGA-II*, *NSGA-III*, and *PO-count*. The corresponding value for *PO-prob* and *PO-prob\** is $gen = 60$. For larger numbers of knapsacks, the value of 100% is reached even faster.

These results demonstrate that compared to the other algorithms, *PO-prob* generates the lowest number of incomparable solutions. Additionally, the fact that *PO-prob* assigns every solution to a separate front indicates its ability to find a clear direction for the evolutionary process. In the next section, we show that this direction also leads to better solutions.

5.1.4. Characteristics: Main Findings

The main findings of this section are the following:

- For two knapsakcs, $n_k = 2$:
  - *PO-prob* performs better in identifying extreme solutions. In the case of random selection, it also results in fragmented coverage of the Pareto-frontier. This tendency can, however, be repaired by using traditional non-dominated sorting during later generations, as in *PO-prob\**.
  - *NSGA-II* performs the best in covering the middle part of the Pareto-frontier. The behavior and performance of *NSGA-III* and *PO-count* are similar to that of *NSGA-II*.
- When $n\_k > 2$, the tendency changes:
  - *PO-prob* does not result in fragmented sets of solutions; this is observed for $n_k \geq 3$ in our experiments.
  - *PO-prob* and *NSGA-III* result in fewer extreme solutions than other algorithms; this is observed for $n_k \geq 4$ in our experiments.
- Ranking based on probabilistic *PO* results in a very low number of incomparable solutions, which is not the case for *PO-count* and *PD* sorting. This demonstrates the ability of the probabilistic *PO* sorting to distinguish between the solutions and find the direction for further evolution.

*5.2. Performance*

To measure the performance of the proposed approaches, we use two metrics: hypervolume and the fraction of solutions dominated by other algorithms.

5.2.1. Hypervolume

Hypervolume is the metric that indicates the volume of the hyperspace dominated by a set of solutions. This metric is widely adopted in practice both for evaluation of genetic optimization algorithms and for guiding the evolutionary process in indicator-based methods [38]. The value of hypervolume depends on the chosen reference point. To simplify the computational process and due to the fact that we solve a maximization problem, we adopt the origin of coordinates as a reference point for hypervolume calculation. In our case, this point also corresponds to the worst possible solution. Larger hypervolume values indicate better performance.

The values of the hypervolume indicator obtained for different numbers of knapsacks for *NSGA-II* are presented in the second column of Table 3. The first part of the table corresponds to random selection and the second to tournament selection. We can see that the hypervolume values obtained with random selection are of the same scale as those obtained with tournament selection. In addition, when $n_k < 10$, random selection results in higher values of hypervolume. However, when the number of knapsacks reaches 10, tournament selection becomes more effective in terms of hypervolume maximization.

Columns 3–6 show the relative increase (positive number, bold font) or decrease (negative number) in the hypervolume indicator for other algorithms. For a more intuitive representation, the values of these columns are also graphically presented in Figure 9. The first observation that we can make is that *NSGA-III*, while developed for many-objective optimization, almost always results in lower hypervolume values, even for larger numbers of objectives. This supports similar observation from [39]. *NSGA-III* reaches its minimum in relative increase in hypervolume of approximately $-10\%$ for $n_k = 7$ for random selection and for $n_k = 8$ for tournament selection. After that, it starts increasing if random selection is used, and it does not change much for tournament selection.

**Table 3.** Increase in hypervolume relative to NSGA-II, %.

| $n_k$ | NSGA-II, h-Vol. | NSGA-III | PO-count | PO-prob | PO-prob* |
|---|---|---|---|---|---|
| | | Random selection | | | |
| 2 | $9.35 \times 10^7$ | −0.40 | −0.19 | 4.36 | 4.48 |
| 3 | $7.98 \times 10^{11}$ | −0.73 | −0.80 | 3.18 | 1.97 |
| 4 | $6.50 \times 10^{15}$ | −1.95 | −0.13 | 2.15 | 1.14 |
| 5 | $5.06 \times 10^{19}$ | −7.11 | −0.15 | −1.36 | −0.90 |
| 6 | $3.99 \times 10^{23}$ | −10.67 | 0.33 | −1.63 | −0.74 |
| 7 | $3.08 \times 10^{27}$ | −12.15 | −0.56 | −1.14 | 0.55 |
| 8 | $2.26 \times 10^{31}$ | −11.45 | 0.23 | 0.76 | 2.53 |
| 10 | $1.21 \times 10^{39}$ | −10.78 | −0.31 | 6.07 | 7.88 |
| 15 | $2.18 \times 10^{58}$ | −4.23 | −0.14 | 24.13 | 25.02 |
| 25 | $5.73 \times 10^{96}$ | −0.04 | 0.07 | 61.23 | 51.88 |
| | | Tournament selection | | | |
| 2 | $9.01 \times 10^{07}$ | −0.09 | −0.33 | 1.14 | 0.67 |
| 3 | $7.55 \times 10^{11}$ | 0.12 | −0.29 | 3.11 | 2.66 |
| 4 | $6.12 \times 10^{15}$ | −0.79 | −0.33 | 1.16 | 0.48 |
| 5 | $4.73 \times 10^{19}$ | −3.49 | 0.22 | −1.42 | −0.77 |
| 6 | $3.74 \times 10^{23}$ | −7.18 | 0.24 | −2.13 | −0.54 |
| 7 | $2.88 \times 10^{27}$ | −9.07 | 0.39 | −2.02 | −0.31 |
| 8 | $2.16 \times 10^{31}$ | −10.54 | −0.51 | −2.01 | 0.25 |
| 10 | $1.17 \times 10^{39}$ | −10.02 | 0.20 | 0.90 | 4.35 |
| 15 | $2.21 \times 10^{58}$ | −7.90 | −0.76 | 11.67 | 14.14 |
| 25 | $6.49 \times 10^{96}$ | -11.12 | −1.86 | 30.35 | 28.59 |

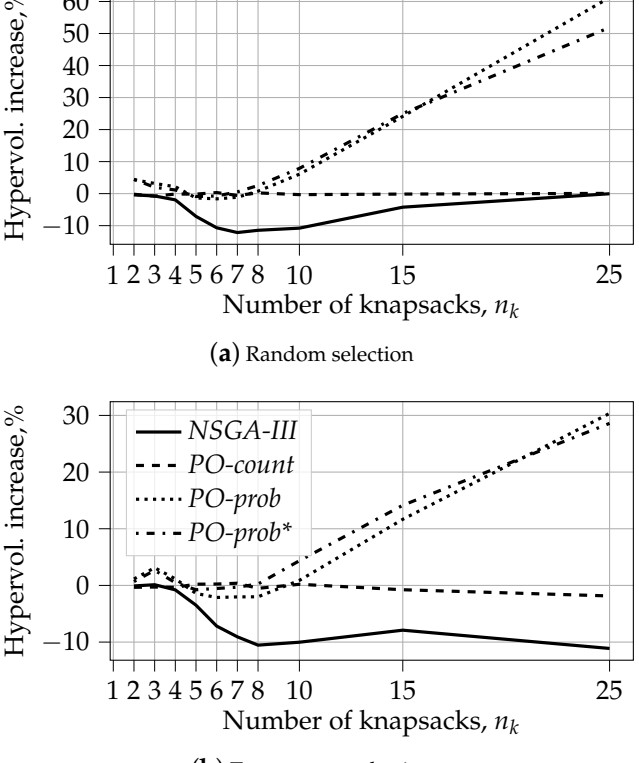

(**a**) Random selection

(**b**) Tournament selection

**Figure 9.** Relative increase in hypervolume compared to *NSGA-II* as a function of the number of knapsacks $n_k$. The legend is shared among plots.

The values of relative hypervolume increase for *PO-count* are very close to 0. This means that *PO-count* results in a population covering the same hypervolume as *NSGA-II*. The largest difference between *NSGA-II* and *PO-count* is observed for $n_k = 25$ with tournament selection. In this case, the hypervolume of *PO-count* is 1.86% less than that of *NSGA-II*. Contrarily, both *PO-prob* and *PO-prob\** improve hypervolume significantly as compared to *NSGA-II*. This difference is visible for small $n_k$ and is especially prominent for large numbers of knapsacks. When $n_k < 5$, *PO-prob* and *PO-prob\** increase the hypervolume by up to 4%. After that, we observe an inverse pattern. However, the relative decrease in this case does not exceed $-2.13\%$. Finally, starting from $n_k = 8$ for random selection and $n_k = 10$ for tournament selection, *PO-prob* again results in larger hypervolume values. For *PO-prob\**, this happens even faster: $n_k = 7$ for random selection and $n_k = 8$ for tournament selection. From this point on, we see a rapid growth with a maximum relative increase observed for $n_k = 25$: $+61.23\%$ and $+51.88\%$ for random selection, and $+30.35\%$ and $+28.59\%$ for tournament selection. The fact that *PO-prob\** is closer to *NSGA-II* than *PO-prob* is due to *PO-prob\** turning into *NSGA-II* during the last 150 generations. We can also notice that the effect of the *PO-prob* algorithm is less visible in the case of tournament selection.

### 5.2.2. Fraction of Dominated Solutions

We calculate the percentage of dominated solutions as follows. For a given pair of algorithms, *algorithm1* and *algorithm2*, we calculate how many solutions of *algorithm2* (*dominated algorithm*) are dominated by solutions of *algorithm1* (*dominating algorithm*). After that, we average the obtained results among all 30 independent runs. The detailed results for $n_k = 2$, $n_k = 7$, and $n_k = 25$ are shown in Tables 4 and 5 for random and tournament selection respectively. For example, for random selection and $n_k = 2$, on average, the solutions of *NSGA-II* dominate 47.38% of the solutions produced by *PO-prob\**, see the third row and sixth column in Table 4. At the same time, on average, only 6.24% of solutions produced by *NSGA-II* are dominated by the solutions of *PO-prob\**, see the seventh row and second column of the same table. Naturally, a better algorithm has a lower number of dominated solutions and a larger number of dominating solutions. This means that we want to achieve minimum per column and maximum per row.

The rows in bold show the average number of dominated solutions for all dominating algorithms, denoted by $\theta$. The value of $\theta$ is calculated as a mean per column, and better algorithms have lower values of $\theta$. For example, for the same setup, on average, 21.77% of solutions produced by *NSGA-II* are dominated by other algorithms. The corresponding value for *PO-prob\** is 36.86%. This means that for this configuration, *NSGA-II* performs better than *PO-prob\**.

**Table 4.** Percentage of dominated solutions for random selection, %.

| | Dominated algorithm, $n_k = 2$ | | | | |
| --- | --- | --- | --- | --- | --- |
| | *NSGA-II* | *NSGA-III* | *PO-count* | *PO-prob* | *PO-prob\** |
| *NSGA-II* | | 44.24 | 39.21 | 13.52 | 47.38 |
| *NSGA-III* | 37.50 | | 34.51 | 13.52 | 44.76 |
| *PO-count* | 43.34 | 45.01 | | 13.52 | 46.94 |
| *PO-prob* | 0.00 | 0.00 | 0.00 | | 8.37 |
| *PO-prob\** | 6.24 | 6.64 | 5.02 | 23.46 | |
| **mean, $\theta$** | **21.77** | **23.97** | **19.60** | **16.01** | **36.86** |
| | Dominated algorithm, $n_k = 7$ | | | | |
| | *NSGA-II* | *NSGA-III* | *PO-count* | *PO-prob* | *PO-prob\** |
| *NSGA-II* | | 0.00 | 8.57 | 0.00 | 0.04 |
| *NSGA-III* | 39.72 | | 40.51 | 0.08 | 19.00 |
| *PO-count* | 9.99 | 0.00 | | 0.00 | 0.05 |
| *PO-prob* | 66.73 | 27.51 | 66.72 | | 48.65 |
| *PO-prob\** | 61.52 | 1.41 | 63.31 | 0.07 | |
| **mean, $\theta$** | **44.49** | **7.23** | **44.78** | **0.04** | **16.94** |

**Table 4.** *Cont.*

| | NSGA-II | NSGA-III | PO-count | PO-prob | PO-prob* |
|---|---|---|---|---|---|
| | Dominated algorithm, $n_k = 25$ | | | | |
| NSGA-II | | 0.00 | 8.28 | 0.00 | 0.15 |
| NSGA-III | 18.97 | | 18.83 | 0.00 | 8.48 |
| PO-count | 6.40 | 0.00 | | 0.00 | 0.20 |
| PO-prob | 36.55 | 9.59 | 36.52 | | 18.76 |
| PO-prob* | 30.13 | 0.80 | 31.89 | 0.00 | |
| mean, $\theta$ | 23.01 | 2.60 | 23.88 | 0.00 | 6.90 |

**Table 5.** Percentage of dominated solutions for tournament selection, %.

| | NSGA-II | NSGA-III | PO-count | PO-prob | PO-prob* |
|---|---|---|---|---|---|
| | Dominated algorithm, $n_k = 2$ | | | | |
| NSGA-II | | 43.93 | 46.67 | 47.81 | 61.20 |
| NSGA-III | 35.99 | | 43.17 | 45.71 | 59.33 |
| PO-count | 35.83 | 36.70 | | 42.32 | 57.43 |
| PO-prob | 6.61 | 7.75 | 7.53 | | 31.29 |
| PO-prob* | 7.78 | 9.95 | 10.41 | 42.26 | |
| mean, $\theta$ | 21.55 | 24.58 | 26.95 | 44.52 | 52.31 |
| | Dominated algorithm, $n_k = 7$ | | | | |
| NSGA-II | | 0.00 | 11.21 | 0.00 | 0.09 |
| NSGA-III | 42.92 | | 44.00 | 0.53 | 15.69 |
| PO-count | 12.16 | 0.00 | | 0.01 | 0.03 |
| PO-prob | 63.32 | 14.72 | 63.73 | | 39.24 |
| PO-prob* | 62.91 | 1.71 | 61.52 | 0.11 | |
| mean, $\theta$ | 45.33 | 4.11 | 45.12 | 0.16 | 13.76 |
| | Dominated algorithm, $n_k = 25$ | | | | |
| NSGA-II | | 0.00 | 6.69 | 0.03 | 0.32 |
| NSGA-III | 17.35 | | 20.83 | 0.03 | 10.60 |
| PO-count | 5.57 | 0.00 | | 0.01 | 0.12 |
| PO-prob | 29.77 | 3.60 | 30.92 | | 20.40 |
| PO-prob* | 25.71 | 0.44 | 27.97 | 0.23 | |
| mean, $\theta$ | 19.60 | 1.01 | 21.60 | 0.07 | 7.86 |

**Results for $n_k = 2$ with random selection.** Analyzing the results from Table 4, we can see that for two knapsacks and random selection, *PO-prob* is dominated the least number of times. *NSGA-II*, *NSGA-III*, and *PO-count*, on average, dominate no more than 13.52% of the solutions of *PO-prob*. *PO-prob\**, however, dominates on average 23.46% of the solutions of *PO-prob*. At the same time, *PO-prob* almost never dominates other algorithms. The only corresponding non-zero value is 8.37%, which represents the fraction of solutions of *PO-prob\** dominated by *PO-prob*. This means that solution spaces of *PO-prob* and other algorithms are distinct and do not intersect much. This supports our previous observation, see Figure 4.

*PO-prob\** is dominated the most often by others, with more than 40% of solutions being dominated by *NSGA-II*, *NSGA-III*, and *PO-count*.

*NSGA-II*, *NSGA-III*, and *PO-count* form a group of algorithms that are not usually dominated by the probability-based algorithms, *PO-prob* and *PO-prob\**, but have high domination values among themselves. Among these three algorithms, *PO-count* seems to provide the best configuration: it is dominated the least number of times by *NSGA-II* and *NSGA-III*; the corresponding values are 39.21% and 34.51%, respectively. At the same time, *PO-count* dominates 43.34% of the solutions of *NSGA-II* and 45.01% of the solutions of *NSGA-III*. This corresponds to +5% and +10% compared to the inverse domination relationship.

**Results for $n_k = 2$ with tournament selection.** For the same number of knapsacks, $n_k = 2$, and tournament selection, we can see that the probability-based algorithms are

dominated more often than the counting-based algorithms, see Table 5. The average number of dominated solutions through all algorithms, $\theta$, is 44.52% for *PO-prob* and 52.31% for *PO-prob\**. At the same time, the values of $\theta$ for *NSGA-II*, *NSGA-III*, and *PO-count* stay relatively close to the corresponding values for random selection.

The results for this configuration, also confirm that *NSGA-II* performs the best: it is dominated the least often and on average dominates more solutions that other algorithms. This difference becomes especially prominent when comparing it with probability based algorithms. *NSGA-II* dominates 47.81% of the solutions of *PO-prob* and 61.20% of solutions of *PO-prob\**. The latter algorithms, however, dominate only 6.61% and 7.78% of the solutions of *NSGA-II*, respectively.

**Results for $n_k = 7$.** However, this pattern changes when the number of objectives increases. Already for $n_k = 7$, *NSGA-II* and *PO-count* are substantially outperformed by other algorithms, both for random and tournament selection. More than 60% of solutions of these two algorithms are dominated by the probability-based algorithms *PO-prob* and *PO-prob\**. The level of domination in the inverse direction is less than 1%.

It is interesting to note that *NSGA-III* also results in a set of solutions that are rarely dominated. The only algorithm capable of dominating a significant fraction of solutions of *NSGA-III* is *PO-prob*. The corresponding values are 27.51% for random selection and 14.72% for tournament selection. This shows the superiority of *NSGA-III* for many-objective optimization as compared to *NSGA-II*. As it was shown in Section 5.2.1, the solutions of *NSGA-III* cover less hypervolume than the solutions of *NSGA-II*. However, as we can see now, approximately 40% of the solutions of *NSGA-II* are dominated by the solutions of *NSGA-III*. Thereby, despite covering less hypervolume, *NSGA-III* should be preferable in practice.

A considerable fraction of solutions of *PO-prob\** is dominated by *NSGA-II* (19.00% and 15.69%) and by *PO-prob* (48.65% and 39.24%). At the same time, only a tiny fraction of solutions of *PO-prob* is dominated by solutions produced by other algorithms. The largest fraction of solutions of *PO-prob* are dominated by *NSGA-III*: 0.08% for random selection and 0.53% for tournament selection.

**Results for $n_k = 25$.** When the number of knapsacks increases even further, we can notice that all algorithms tend to produce more distinct sets of solutions, as the domination fractions reduce. However, the general pattern stays the same. The solutions of *NSGA-II* and *PO-count* are more often dominated by the solutions of other algorithms. The solutions of *PO-prob* are almost never dominated. The next best performance is demonstrated by *NSGA-III*, with 9.59% and 3.60% of solutions dominated by *PO-prob* for random and tournament selection, respectively. *PO-prob\** has approximately 20% of solutions dominated by *PO-prob* and 10% of solutions dominated by *NSGA-III*.

**Distribution of $\theta$.** To further analyze how the domination fraction changes for different numbers of knapsacks, we demonstrate the distribution of $\theta$ for different values of $n_k$ in Figure 10. Recall that $\theta$ shows the fraction of dominated solutions averaged over different dominating algorithms, and its values are present in bold in Tables 4 and 5.

We can notice similar tendencies for both random and tournament selection. *NSGA-II* and *PO-count* behave very similarly. For $n_k = 2$, the value of $\theta$ for these algorithms is around 23%. After that, it starts increasing and reaches its peak of approximately 45% for $n_k = 7$. Finally, it gradually decreases to 20% for $n_k = 25$.

*NSGA-III* starts at a similar level. It reaches its peak of approximately 30% for $n_k = 5$ for random selection and $n_k = 3$ for tournament selection. After, it decreases below 10% for $n_k = 8$ and stays below this value for random selection, and relatively close to zero in the case of tournament selection. These results once again demonstrate the superiority of *NSGA-III* over *NSGA-II* for large numbers of objectives.

*PO-prob* starts at around 15% for random selection and 45% for tournament selection. However, the value of $\theta$ drops to 0 very fast. This shows that the solutions produced by this algorithm are almost never dominated by any other algorithm for large numbers of knapsacks.

*PO-prob\** behaves similarly to *PO-prob* until $n_k = 5$. Next, $\theta$ starts going up and reaches its maximum at $n_k = 7$ for random selection and at $n_k = 8$ for tournament selection. For larger values of $n_k$, $\theta$ gradually decreases, but it is always larger than the corresponding value for *NSGA-III*.

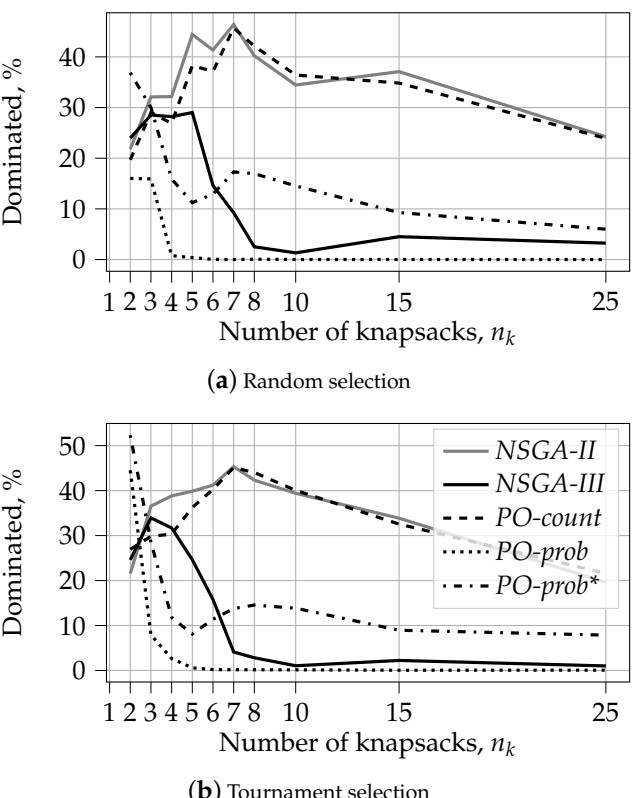

**(a)** Random selection

**(b)** Tournament selection

**Figure 10.** Average percentage of solutions dominated by other algorithms. The legend is shared among plots.

### 5.2.3. Performance: Main Findings

In general, the results presented in this subsection demonstrate the following:

- The performance of *PO-count* is very close to that of *NSGA-II*, both in terms of hypervolume and the fraction of dominated solutions.
- Our results support the finding from [39]. We show that, contrary to expectations, *NSGA-III* results in lower values of hypervolume than *NSGA-II* for large numbers of objectives. However, we also show that for large values of $n_k$, the solutions produced by *NSGA-III* are rarely dominated by those produced by *NSGA-II*. At the same time, *NSGA-III* does dominate some fraction of solutions produced by *NSGA-II*. This shows that *NSGA-III* can be beneficial for many-objective optimization problems. This observation was not reported in [39].
- Finally, our experiments clearly demonstrate the advantages of probability-based algorithms for many-objective optimization problems. Both *PO-prob* and *PO-prob\** result in solutions that cover larger hypervolumes and are less often dominated by the solutions of other algorithms.

### 5.3. Time Complexity

In this section, we analyze the time complexity of the studied algorithms. The proposed algorithms differ from *NSGA-II* only in the sorting method used when selecting the best elements for the next generation. In addition, this is the most computationally expensive part. The execution times of the other parts of the algorithm are proportional to the population size. That is why we report the duration of the sorting procedure for the studied algo-

rithms in Figure 11. This figure shows the results for 10 knapsacks and random selection. The general trends and relations are similar for all other configurations.

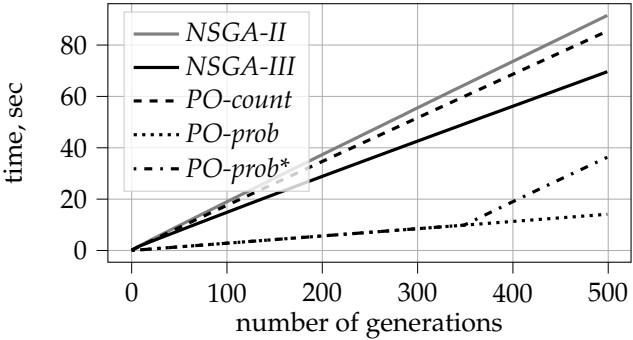

(**a**) Cumulative sorting duration as a function of number of generations

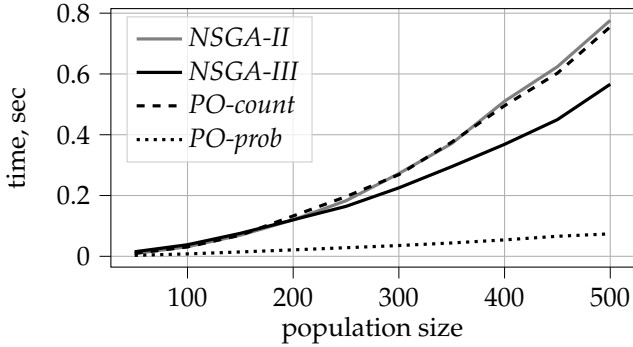

(**b**) Sorting duration as a function of population size, *pop_size*

**Figure 11.** Time complexity for random selection and 10 knapsacks, $n_k = 10$.

**Cumulative sorting duration.** Figure 11a shows cumulative sorting duration as a function of the number of generations. As the size of the population remains the same for every generation, cumulative sorting duration increases linearly for all algorithms except *PO-prob\**. This algorithm is a combination of *PO-prob* for generations 0–350 and *NSGA-II* afterwards. That is why for *PO-prob\** we observe a curve with two linear pieces with corresponding inclinations.

We can also observe that *NSGA-II* and *PO-count* require similar time for sorting, and *NSGA-III* performs the sorting procedure slightly faster. Contrarily, time complexity of *PO-prob* and *PO-prob\** is significantly less. Indeed, *PO-prob* does not perform pairwise comparison of elements in the current population. Instead the values are compared separately for every objective and are afterwards multiplied to obtain approximate ranking.

These experimental results confirm theoretical expectations based on the definition of the proposed algorithm.

**Sorting duration as a function of** *pop_size*. In Figure 11b, we report the dependency of sorting time on the population size for values of *pop_size* ranging from 50 to 500. The reported values are averages of 100 independent executions of one iteration of the corresponding genetic algorithm. We do not show the results for *PO-prob\** here, as it is an aggregation of two other algorithms.

From the figure, we can see that *PO-prob* requires much less time than all other algorithms. The results for *NSGA-II* and *PO-count* tend to be very close, as in other experiments. Finally, *NSGA-III* is in between.

This observation also has a theoretical explanation. Indeed, choosing the next generation for *NSGA-III* and *NSGA-II* has time complexity of $max\{O(N^2M), O(N^2 log_{M-2}N)\}$ and $O(N^2M)$ respectively, where $M$ stands for the number of objectives and $N$ is the popu-

lation size, see [6,16]. At the same time, sorting in *PO-prob* comes down to independent sorting procedures with respect to every objective. The time complexity of this procedure is $O(NMlog(N))$.

All these results prove the computational efficiency of the approximate ranking calculation procedure used in *PO-prob* both theoretically and via the experiments.

## 6. Conclusions

In this paper, we study the application of a novel sorting technique based on Pareto optimatily to genetic optimization. The proposed approach is very flexible and can be defined for various sets of elements, relations, and measures. In the case of genetic optimization, the set of elements is represented by a discrete set of points on a plane with the traditional numerical relationship $\geq$. However, we consider two different measures: counting and probability, which result in two versions of the algorithm: *PO-count* and *PO-prob*. The proposed ranking method was implemented in the framework of the *NSGA-II* algorithm and was tested on the 0/1 knapsack problem with the number of objectives $n_k$ ranging from 2 to 25.

Our experimental results demonstrated that counting *PO* performs very similar to the traditional non-dominated sorting. At the same time, probabilistic *PO* ranking has multiple advantages. It allows the genetic algorithm to distinguish better between the solutions, which is particularly useful in the case of many-objective optimization. It also results in higher values of hypervolume both for large and small number of knapsacks as compared to *NSGA-II* and *NSGA-III*. In the case of random selection, hypervolume is increased by 4% for 2 knapsacks and by 61% for 25 knapsacks. The respective values for tournament selection are 1% and 30%. For medium numbers of knapsacks, we observe a decrease in hypervolume, but it stays within $-2\%$. Apart from increasing the hypervolume when the number of objectives becomes larger, *PO-prob* also results in a set of solutions that are very rarely dominated by other algorithms. For example, for 10 knapsacks, none of the solutions of *PO-prob* are dominated by *NSGA-II*. At the same time, *PO-prob* dominates approximately 60% of the solutions produced by *NSGA-II*. Finally, we demonstrated both theoretically and practically the efficiency of *PO-prob* in terms of execution time. Sorting based on probabilistic *PO* allows decreasing the computational time from $O(N^2M)$ to $O(NMlog(N))$, where $N$ and $M$ stand for populations size and number of objectives, respectively. The fact that *PO-prob* performs better than *PO-count* indicates that the practical problem formalization might be crucial. In this case, the choice of the probability measure over the counting one improves the performance drastically.

Although *PO-prob*-based sorting demonstrated good performance, this approach has some limitations. It was demonstrated that it introduces front fragmentation for $n_k = 2$ and slightly decreases hypervolume when the number of objectives is between five and eight. These problems can be partially solved by the *PO-prob\** algorithm, which is a combination of *PO-prob* and the traditional *PD* ranking used in the original version of *NSGA-II*. However, when the number of objectives is large, *PO-prob* provides the best performance.

As it was experimentally demonstrated in Section 5.1.3, sorting by *PO-prob* results in almost no ties. In this sense, it can be considered as a linear extension of non-dominated sorting. However, compared to other popular linear extensions, such as sum, average, minimum, or maximum of objectives, *PO-prob* has a clear **interpretation**. It represents the probability that a randomly chosen point will dominate the point under investigation, see Section 3. Thereby, the resulting algorithms are easier to manage and understand.

## 7. Future Work

There are several experimental evaluations that we would like to perform as future work. First, we would like to study the effect of correlated objectives. It is known from the literature that *NSGA-II* performs better in the case of correlated objectives [40]. We want to study the behavior of *PO*-based algorithms in this case. Next, we would like to test the performance of our algorithms in the framework of other optimization problems, including those extracted from real-world optimization applications, for example, those

from [41]. It might be also interesting to compare our algorithms with the following approaches and frameworks: other number-based methods of dominance re-definitions, see Section 2.1; approximate non-dominate sorting [30,32]; extreme non-dominated sorting, which utilizes extreme solutions in the population generation in order to enhance the quality of solutions [42,43]; approaches based on a subset of Pareto optimal solutions for which an improvement in one objective will result in a severe degradation in at least another one [9]; and Dominance Resistant Solutions [44], which have very good values for some objectives and very bad values for other objectives.

We also envision several directions for further theoretical analysis of the proposed approach. We would like to study the problem of front splitting that was observed for $n\_k = 2$. As suggested in Section 5.1.1, employing an adaptive parents selection strategy can provide a solution. Additionally, we want to analyze the effect of constraints violation and the application of the proposed methods to constrained genetic optimization. Indeed, constraints can be formulated as additional objectives resulting in the increase in the problem's dimensionality. However, as it was demonstrated in this paper, *PO-prob* can effectively cope with such problems.

Moreover the framework proposed in [14] offers a lot of flexibility. Using appropriate relations and measures, *k*-Pareto optimality might be adapted to select the simplest solutions, for example using a sub-tree or a sub-graph relation; the best solutions of mixed-integer objectives; or to express trade-offs using cone-based relations.

**Author Contributions:** Conceptualization, J.R.; methodology, J.R.; software, M.A.; validation, J.R. and M.A.; formal analysis, J.R.; investigation, J.R. and M.A.; writing—original draft preparation, M.A.; writing—review and editing, J.R., M.A. and T.E.; visualization, J.R. and M.A.; supervision, T.E. All authors have read and agreed to the published version of the manuscript.

**Funding:** This research received no external funding.

**Data Availability Statement:** Not applicable.

**Conflicts of Interest:** The authors declare no conflict of interest.

**Abbreviations**

The following abbreviations are used in this manuscript:

| | |
|---|---|
| NSGA-II | Non-dominated Sorting Genetic Algorithm II [6] |
| NSGA-III | Non-dominated Sorting Genetic Algorithm III [16] |
| PD | Pareto dominance-based sorting, as used in NSGA-II |
| PESA-II | Pareto Envelope-based Selection Algorithm II [5] |
| (k-)PO | (k-)Pareto Optimality [14] |
| PO-count | Pareto Optimality computed via counting |
| PO-prob | Pareto Optimality computed using probabilistic approximation |
| PO-prob* | PO-prob sequentially combined with Pareto dominance sorting |
| SPEA2 | Strength Pareto Evolutionary Algorithm 2 [7] |

**Appendix A. 0/1 Multi-objective Knapsack Problem Formulation**

The 0/1 multi-objective knapsack problem from [35] is formulated in the following way. Let $n$ be the number of knapsacks and $m$ the number of items. Let $p_{i,j}$ be the profit of item $j$ according to knapsack $i$, $w_{i,j}$ the weight of item $j$ according to knapsack $i$, and $c_i$ the capacity of knapsack $i$. The problem is to find a vector $\vec{x} = (x_1, x_2, \ldots, x_m) \in \{0,1\}^m$ such that

$$\forall i \in \{1, 2, \ldots, n\} : \sum_{j=1}^{m} w_{i,j} x_j \leq c_i \tag{A1}$$

and for which $\vec{f}(\vec{x}) = (f_1(\vec{x}), f_2(\vec{x}), \ldots, f_n(\vec{x}))$ is maximum, where

$$f_i(\vec{x}) = \sum_{j=1}^{m} p_{i,j} x_j \tag{A2}$$

and $x_j = 1$ iff item $j$ is selected.

We choose uncorrelated profits $p_{i,j}$ and weights $w_{i,j}$ as random integers in the interval $[10, 100]$. The knapsack capacities are set to half of the total of the corresponding item weights:

$$c_i = 0.5 \sum_{j=1}^{m} w_{i,j}. \tag{A3}$$

A particular vector $\vec{x}$ generated randomly or via a genetic procedure can violate the capacity restriction of Equation (A1) for a knapsack. In this case, following [35], we adopt a greedy repair method. This method removes items from a particular solution until all capacity constraints are fulfilled. The items are removed in the ascending order of maximum profit/weight ratio $q_i$, which is given by the following equation:

$$q_j = \max_{i=1}^{n} \left\{ \frac{p_{i,j}}{w_{i,j}} \right\}. \tag{A4}$$

In this way, the items with lower profit per weight unit $q_j$ are removed first. The aim of this procedure is to fulfill all capacity constraints while reducing the overall profit as little as possible.

## Appendix B. Visualization of the First front with Parallel Coordinates for $n\_k = 2$

Figures A1 and A2 show the visualization of solutions using parallel coordinates for $n\_k = 2$. The horizontal axis of each plot represents the index of the objective, and the vertical axis shows the corresponding objective value. The visualization strategy corresponds to the one described in Section 5.1.2.

The fragmentation of the first front for *PO-prob* with random selection is depicted in Figure A1d. The absence of similar behavior in Figures 6d and 7d is an additional indicator of the fact that in larger dimensions, the solutions of *PO-prob* are not fragmented. In the case of two knapsacks, this tendency is repaired by *PO-prob\**, see Figure A1e. The latter approach also covers a larger range of objective values than *NSGA-II*, see Figure A1a.

The fragmentation is not observed in the case of tournament selection, see Figure A2d. Additionally, we see that, in this case, one objective is preferred over another. This is clearly depicted in the absence of symmetry in Figures A2a,d,e.

All these observations support the findings from Section 5.1.1.

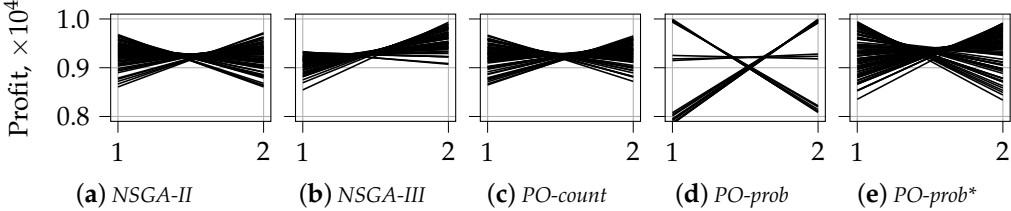

**(a)** *NSGA-II*    **(b)** *NSGA-III*    **(c)** *PO-count*    **(d)** *PO-prob*    **(e)** *PO-prob\**

**Figure A1.** Profit of the knapsacks from the final front for $n_k = 2$ with random selection. Different knapsacks are encoded by numbers on the *x*-axis. *y*-axis is shared.

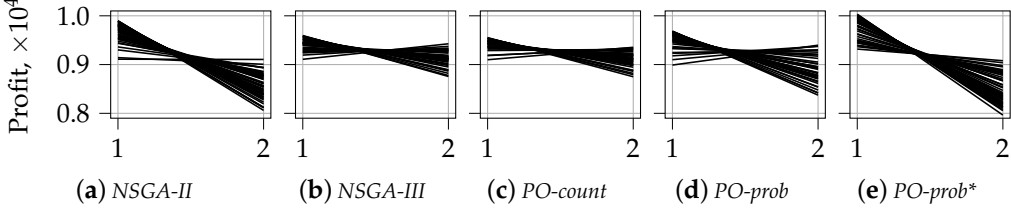

**(a)** *NSGA-II*    **(b)** *NSGA-III*    **(c)** *PO-count*    **(d)** *PO-prob*    **(e)** *PO-prob\**

**Figure A2.** Profit of the knapsacks from the final front for $n_k = 2$ with tournament selection, $*10^4$. Different knapsacks are encoded by numbers on the *x*-axis. *y*-axis is shared.

## Appendix C. Average Distance to Diagonal

A diagonal in the $n$-dimensional hyperspace is defined with the following equation: $x_1 = x_2 = \cdots = x_n$. In the case of $n = 2$ and $x$ and $y$ being two axes, the conventional definition of the diagonal is the following: $y = x$. The optimization problem considered in this paper is symmetric. This means that all objectives have the same scale, or ,for all objectives, the values are distributed similarly. In such a case, the Euclidean distance between a solution and the diagonal shows how close (for large distances) or how far (for small distances) the solution is to the extremes. The values of the average distance to the diagonal for four different numbers of knapsacks, $n_k \in \{2, 4, 6, 8\}$, are shown in Figure A3.

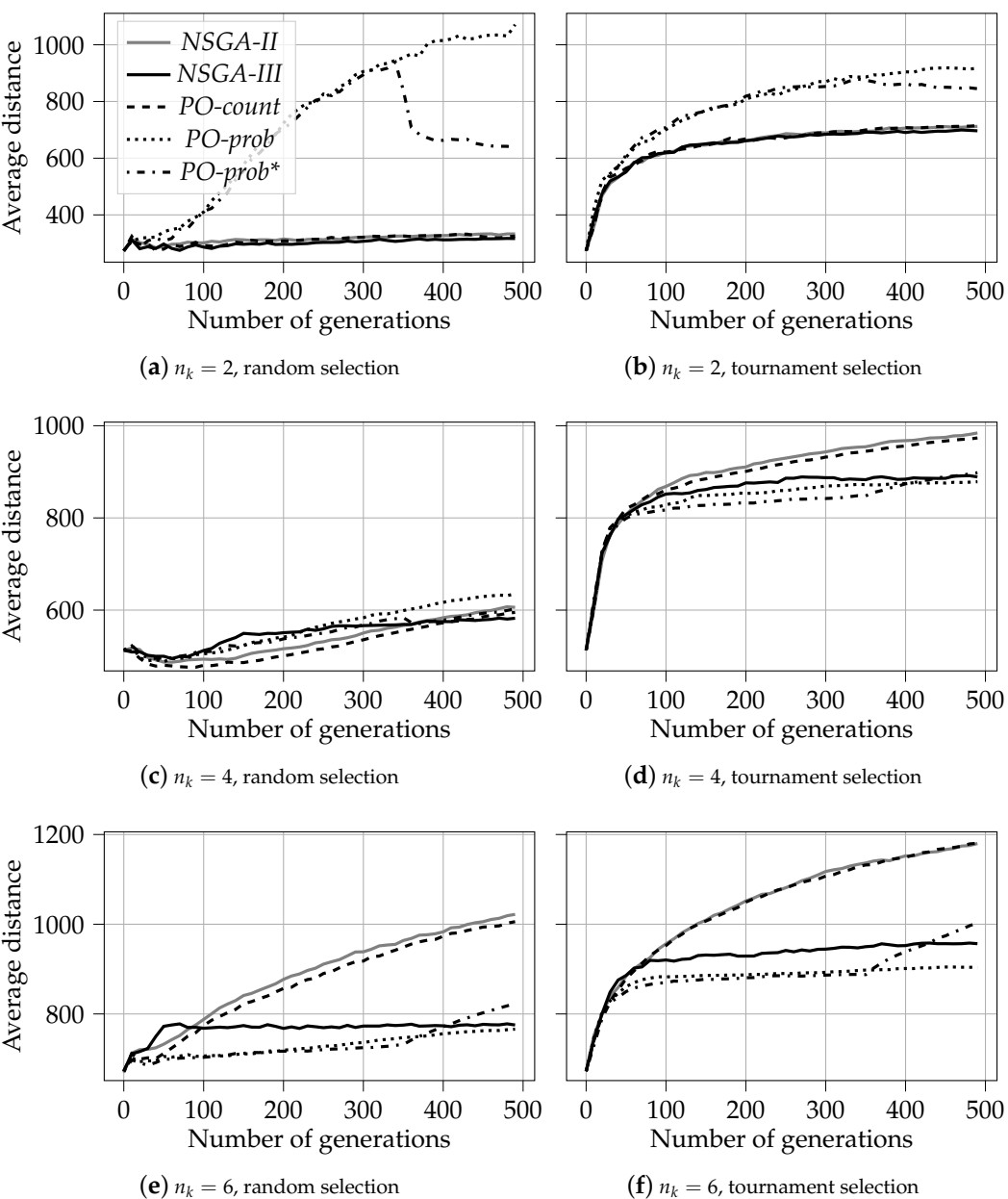

(**a**) $n_k = 2$, random selection

(**b**) $n_k = 2$, tournament selection

(**c**) $n_k = 4$, random selection

(**d**) $n_k = 4$, tournament selection

(**e**) $n_k = 6$, random selection

(**f**) $n_k = 6$, tournament selection

**Figure A3.** *Cont.*

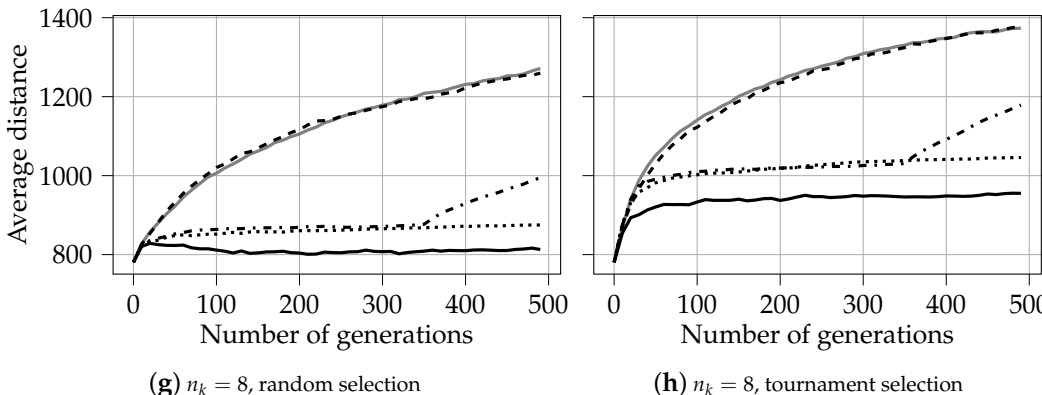

(**g**) $n_k = 8$, random selection        (**h**) $n_k = 8$, tournament selection

**Figure A3.** Average distance to diagonal for different number of knapsacks as a function number of generations. The legend and *y*-axis are shared among plots.

As we can see, on all figures, the values for *PO-prob* and *PO-prob\** are very close to each other up to generation 350. After that, we can observe different behaviors of plots for these two algorithms. This observation, similarly to previous ones, is explained by the definition of *PO-prob\** as a combination of *PO-prob* and *NSGA-II*.

Let us analyze the results for **random selection**. In Figure A3a, we can see that for two knapsacks, $n_k = 2$, the curves for *NSGA-II*, *NSGA-III*, and *PO-count* are very close to each other, and the associated solutions are relatively close to the diagonal. On the other hand, for *gen* $\geq$ 200, the solutions produced by *PO-prob* and *PO-prob\** are more than two times farther away from the diagonal. It means that these algorithms produce solutions that are much closer to extreme values and cover less the central part of the Pareto-frontier. This observation also agrees with the results presented in Section 5.1.1, see Figure 4b. As expected, after the number of generations reaches 350, the solutions of *PO-prob\** tend to be closer to the diagonal, as the *NSGA-II* selection procedure is used.

We can notice, however, that this general trend changes when the number of knapsacks increases. For $n_k = 4$, all algorithms have similar values for the average distance to the diagonal, see Figure A3c. For six knapsacks, the relation is already inverted: the solutions produced by *NSGA-II* and *PO-count* tend to extremes more than those for other algorithms, see Figure A3e. For $n_k = 8$, starting from *gen* = 200, the solutions produced by *PO-prob* are at least 25% closer to the diagonal than those produced by *NSGA-II* and *PO-count*, see Figure 4g. Also note that the solutions produced by *NSGA-III* always have a relatively low distance to the diagonal, regardless of the number of knapsacks. This can be explained by the fact that all chosen solutions in *NSGA-III* are attached to reference points that do not change during the whole evolutionary process. All this supports the relative findings from Section 5.1.2.

In the case of **tournament selection**, we observe a similar dependency between the average distance to the diagonal and the number of knapsacks. The average distance to the diagonal for *PO-prob* and *PO-prob\** decreases when *n_k* increases. This effect becomes visible faster than for random selection; compare Figures A3d,c. Moreover, we can notice that for all algorithms, tournament selection tends to produce solutions that are farther away from the diagonal for larger numbers of knapsacks. In the case of $n_k = 2$, however, *PO-prob* with random selection produces more extreme solutions than with tournament selection, see Figures A3a,b. This observation also corresponds to the results presented in Figures 4b and 5f of Section 5.1.1.

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
