# Peer review of "k-Pareto Optimality-Based Sorting with Maximization of Choice and Its Application to Genetic Optimization"

_algorithms, doi:10.3390/a15110420_

Round 1
Reviewer 1 Report
In regards to the manuscript “K-pareto optimality-based sorting with maximization choice and its application to genetic optimization” where the authors present a redefined calculation of Pareto dominance, where the auras propose to rank solutions according o the measure of dominating solutions and argue that the complexity is improved. The paper is in general well written and this reviewer recommend this work for its publication. However, some points are suggested to be attended before its publication.
- It is suggested to include a flow diagrama for clarity to the algorithm.
- Some tipos in line 205.
- It is suggested to writte explicity measure of dominating solutions employed.
- Why does the values of the third front are spreader into extreme values for the PO-prob? More discussion about this point is suggested.
Author Response
Dear reviewer, we would like to thank you for your time and effort spent on working with our paper. Please find below our answers to the raised comments.
- It is suggested to include a flow diagrama for clarity to the algorithm.
We added a flow chart and its discussion in Section 3.1 - Some tipos in line 205.
The typos were corrected. - It is suggested to writte explicity measure of dominating solutions employed.
We provided a proper mathematical formulation of the proposed approach and the related concepts in Section 3.2. - Why does the values of the third front are spreader into extreme values for the PO-prob? More discussion about this point is suggested.
We added more discussion about this point, see text in red on pages 10 and 11.
Reviewer 2 Report
This paper's main proposal is K-Pareto Optimality-based sorting and optimization with maximization of choice in different genetic algorithms. The approach is described in detail and supported by an experimental setup that is sound in terms of sorting duration, number of non-dominated solutions and time complexity. The paper is well researched but need some improvements:
- Evaluation of solution's quality. Authors should evaluate the quality of solution and should demonstrate the effect of constraint violation on existing genetic optimization algorithms using PO-count and PO-Prob.
- PO-Probability adoption. How and why PO-prob performs better in identifying extreme solutions than original NSGA-II
- Lessons learned. Please could you give some limitations of proposed PO-based individual selection. Besides, add some lessons learned about PO-count, PO-prob and PO-prob* which one best strategy to a particular genetic algorithm.
- Please which means the symbol * at the end of PO-prob
- Future works. Please add some prospects for future research.
Author Response
Dear reviewer, we would like to thank you for your time and effort spent on working with our paper. Please find below our answers to the raised comments.
- Evaluation of solution's quality. Authors should evaluate the quality of solution and should demonstrate the effect of constraint violation on existing genetic optimization algorithms using PO-count and PO-Prob. The quality metrics used in this paper are hypervolume and fraction of dominated solutions. Unfortunately, due to time limitations (10 days to answer the review questions), we are unable to evaluate the quality using other metrics and study the effect of constraint violation. We aim, however, to do it in future work.
- PO-Probability adoption. How and why PO-prob performs better in identifying extreme solutions than original NSGA-II
We added more discussion about this point, see text in red on pages 10 and 11 - Lessons learned. Please could you give some limitations of proposed PO-based individual selection. Besides, add some lessons learned about PO-count, PO-prob and PO-prob* which one best strategy to a particular genetic algorithm.
We added more discussion about limitations of the proposed approaches and about when which approach should be preferred, see text in red in Section 6 Conclusions. Unfortunately, we cannot give general recommendations at this stage, as the experimental results were performed using only 1 test problem. We aim to study this question further in our future work. - Please which means the symbol * at the end of PO-prob
PO-prob* is a modification of PO-prob. This algorithm consists in sequential usage of sorting by k-Pareto optimality and by Pareto dominance. We added footnote 1 on page 2 to guide the reader to detailed explanations from the beginning of the paper and restructured the presentation of different algorithms, see text in red on page 2. - Future works. Please add some prospects for future research.
We framed all future works as a separate section 7 and added more ideas.
Reviewer 3 Report
The paper under consideration deals with the multicriterial (multiobjective) optimization problems being mathematical models of intelligent decision-making processes with several objectives. As a rule, the objectives are contradictory and in the classical approach the solution is defined on the base of a domination principle that generates the Pareto set as the solution to initial multicriterial problem.
The simplest way to find the Pareto set is to build it comparing vectors of objectives according to a dominance relation. This approach is easily implemented, but its efficiency is crucially deteriorated when growing in the problem the number of independent parameters and number of objectives. The traditional attempts to increase efficiency on this way were connected with improving the sorting procedure. The authors take a different approach. They redefine the notion of dominance simplifying it and, as a consequence, reducing complexity of the procedure. Further, the authors add to the procedure elements of stochastics and built these improvements in a version of genetic algorithm.
This version is verified in the computational experiment with examples constructed on the base of 0-1 knapsack problems. Efficiency of the proposed method is compared to 2 known genetic algorithms NSGA-II and NSGA-III. Results of experiments demonstrate the workability of the propose method and its modifications and superiority over NSGA-II and NSGA-III.
Unfortunately, the text of the paper does not stand up to criticism from the scientific point of view. The authors discuss the multicriterial problem, but formulate nowhere its mathematical statement. Additionally, the key definition of k-Pareto optimality has not been introduced. May be, the authors assume that all of humanity knows what is it. And, of course, like a cherry on a cake, the paper submitted to Algorithms contains no description of the proposed method in a general algorithmic form, for instance, as procedure of a pseudo language or, at least, as a flowchart. Instead, operation of the method is explained using a primitive example. Finally, the test problems have not been described in mathematical statements.
The paper must be reconstructed significantly.
Author Response
Dear reviewer, we would like to thank you for your time and effort spent on working with our paper. Please find below our answers to the raised comments.
- The authors discuss the multicriterial problem, but formulate nowhere its mathematical statement.
We added mathematical formulation of a multicriterial optimization problem in Section 3.1. - Additionally, the key definition of k-Pareto optimality has not been introduced.
We added a proper mathematical formulation of the proposed approach and the related concepts in Section 3.2. - the paper submitted to Algorithms contains no description of the proposed method in a general algorithmic form, for instance, as procedure of a pseudo language or, at least, as a flowchart.
We added a flow chart and its discussion in Section 3.1. This flow chart is applicable to both NSGA-II and the methods we propose. As the proposed method consists only in changing the sorting approach of the existing NSGA-II algorithm, we decided not to further describe it in algorithmic form and point the reader to the relevant literature instead. - The test problems have not been described in mathematical statements.
The test problem is described in mathematical statements in Appendix A. Given that the definition of the problem is standard and was described in many research papers, we decided not to present it in the main part of the paper. We added the following phrase to the first paragraph of Section 4 to guide the reader to the formal definitions. “The mathematical formulation of this problem is presented in Appendix A.” - The paper must be reconstructed significantly.
We hope that the significant reconstruction of Section 3 improves the quality of the paper.
Reviewer 4 Report
These are my opinions:
1. There aren't enough citations in the opening section. A thorough explanation of the topic and how it applies to diverse platforms including IoT, Edge, and fog computing is also necessary. The authors encouraged the use of these important references as well.
https://onlinelibrary.wiley.com/doi/abs/10.1002/cpe.7252
https://www.mdpi.com/2076-3417/12/16/8232
https://www.mdpi.com/2076-3417/12/17/8906
https://www.sciencedirect.com/science/article/pii/S131915782200101X
2-Donot leave any sections empty. for instance, the authors abruptly transitioned from section 2 to 2. 1. Please provide a thorough description in section 2.
3-The suggested approach and outcomes must be explained in detail by the authors. These versions are insufficient.
4-The English language must be enhanced.
5-The conclusion section is too lengthy and should be cut. The method's disadvantages must also be mentioned.
Author Response
Dear reviewer, we would like to thank you for your time and effort spent on working with our paper. Please find below our answers to the raised comments.
- There aren't enough citations in the opening section.
As it was suggested, we added more citations about possible applications of genetic optimization, see text in red on page 1. - Do not leave any sections empty. For instance, the authors abruptly transitioned from section 2 to 2.1. Please provide a thorough description in section 2.
We added a description of section 2. - The suggested approach and outcomes must be explained in detail by the authors. These versions are insufficient.
We added more details to the description of the approach in Section 3. Additionally, more analysis is provided in Section 6 Conclusions. - The English language must be enhanced.
We revised the text of the paper. - The conclusion section is too lengthy and should be cut. The method's disadvantages must also be mentioned.
We have split the final section into Section 6 Conclusions and Section 7 Future Work. We hope this improves the readability of the text. We also added the discussion about the disadvantages of the methods.
Round 2
Reviewer 3 Report
All needed corrections taking into account the remarks made in the reviewer's report have been done. May be, the section dealing with experimental results may be presented in a more concise form, but this is just a recommendation.
Reviewer 4 Report
I see no significant improvements in the paper.